# RINGING RELUS: HARMONIC DISTORTION ANALYSIS OF NONLINEAR FEEDFORWARD NETWORKS

**Christian H.X. Ali Mehmeti-Göpel**
Institute of Computer Science
Johannes-Gutenberg University Mainz
Staudingerweg 9, 55122 Mainz, Germany
chalimeh@uni-mainz.de

**David Hartmann**
Institute of Computer Science
Johannes Gutenberg-University of Mainz
Staudingerweg 9, 55128 Mainz, Germany
dahartma@uni-mainz.de

**Michael Wand**
Institute of Computer Science
Johannes Gutenberg-University of Mainz
Staudingerweg 9, 55128 Mainz, Germany
mwand@uni-mainz.de

## ABSTRACT

In this paper, we apply harmonic distortion analysis to understand the effect of nonlinearities in the spectral domain. Each nonlinear layer creates higher-frequency harmonics, which we call "blueshift", whose magnitude increases with network depth, thereby increasing the "roughness" of the output landscape. Unlike differential models (such as vanishing gradients, sharpness), this provides a more global view of how network architectures behave across larger areas of their parameter domain. For example, the model predicts that residual connections are able to counter the effect by dampening corresponding higher frequency modes. We empirically verify the connection between blueshift and architectural choices, and provide evidence for a connection with trainability.

## 1 INTRODUCTION

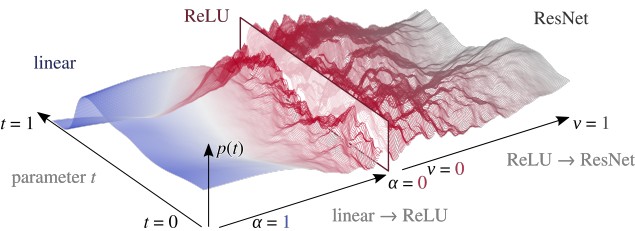

Figure 1: Continous transition of a loss path between linear feedforward ("linear"), nonlinear feedforward ("ReLU") and nonlinear residual ("ResNet") regimes. Graph: loss path near initialization of a ResNet56 v2 with LReLUs with negative slope $\alpha \in [0, 1]$ and residual branch weight $\nu \in [0, 1]$. Left: $\alpha = 0, \nu = 0$, Middle: $\alpha = 1, \nu = 0$, Right: $\alpha = 1, \nu = 1$.

In the past decade, the emergence of practical deep neural networks arguably has had disruptive impact on applications of machine learning. Depth as such appears to be key to expressive models (Raghu et al., 2017). However, depth also comes with challenges concerning training stability. Theoretical problems include vanishing and exploding gradients (Hochreiter, 1991), chaotic feed-forward dynamics (Poole et al., 2016), or decorrelation of gradients (Balduzzi et al., 2017). In practice, a number of "recipes" are widely used, such as specific nonlinearities (Glorot et al., 2011; He et al., 2015), normalization methods such as batch normalization (Ioffe & Szegedy, 2015), shortcut architectures (Srivastava et al., 2015; He et al., 2016a;b), or multi-path architecture with (Huang et al., 2017) and without shortcuts (Szegedy et al., 2016). Broadly speaking, a key research question is to understand how the shape of the network function, i.e., the map from inputs and parameters to outputs, is affected by architectural choices.

Our paper considers specifically the *roughness* of the *weights-to-outputs function* ("*w-o function*") of nonlinear feed-forward networks. Motivated by the recent visualizations of (Li et al., 2018), which show how depth increases roughness and residual connections smoothen the output again, our goal is to provide an analytical explanation of this effect, and study its implications on network design and trainability. To this end, we first formalize "roughness" as the decay-rate of the expected power spectrum of a function class. Our main contribution is to then apply *harmonic distortion analysis* to nonlinear feedforward networks, which predicts the creation of high-frequency "harmonics" (thereby "blueshifting" the power spectrum) by polynomial nonlinearities with large higher-order coefficients. Based on this model, we discuss how network depth increases blueshift and thus roughness, while shortcut connections, low-degree nonlinearities and parallel computation paths dampen it. In relation to trainability, we show an analytic link between blueshift and exploding gradients. Unlike the former model, the spectral view describes a more global behavior of the w-o function over regions in the parameter domain.

Experiments confirm the theoretical predictions: We observe the predicted effects of depth, shortcuts and parallel computation on blueshift, and are able to differentiate different types of nonlinearities by the decay rate of coefficients of a polynomial approximation. The findings are in-line with known advantages in trainability of the different architectures. We further strengthen the evidence by training a large set of networks with a different amount of nonlinearity and depth, which shows a clear correlation between blueshift and training-problems, as well as a trade-off with expressivity.

In summary, our paper explains how network architecture affects roughness, shows a connection to trainability, and thereby provides a new tool for analyzing the design of deep networks.

## 2    RELATED WORK

Vanishing or exploding gradients are a central numerical problem (Hochreiter, 1991; Pascanu et al., 2013; Yang et al., 2019): If the the magnitudes of the singular values of layer Jacobians deviates from one, subspaces are attenuated ($|\sigma| < 1$) or amplified ($|\sigma| > 1$), potentially cascading exponentially over multiple layers (Pennington et al., 2017a). Formally, the behavior of stacks of matrices and nonlinear functions can be modeled by random matrix theory or Gaussian mean-field approximations (Poole et al., 2016; Pennington et al., 2017b; 2018). The gist is that at initialization, orthogonal weight matrices are needed, which is challenging for convolutional architectures. A solution for $\tanh$-networks is given by Xiao et al. (2018); for $\mathrm{ReLU}$, there is a negative result (Pennington et al., 2017b). Using mean-field theory, it can be shown that batch normalization (Ioffe & Szegedy, 2015) leads to exploding gradients at initialization (Yang et al., 2019) (which equalize after a few steps, but that might be too late (Frankle et al., 2020)).

A different route is taken by Balduzzi et al. (2017), who observe an increasing decorrelation of gradients in the input space. Similar to our paper, they show that deeper networks lead to spectral whitening (starting from brown noise); however, the analysis is performed with respect to the inputs $\mathbf{x}$, not weights $\mathbf{W}$. The scale-space structure shown by our model might give further hints on the mechanisms behind training difficulties.

By visualizing random slices of the loss surface, Li et al. (2018) observe that the loss surface of deep feedforward networks transitions between nearly convex to chaotic with increasing depth; our work explains these observations by spectral analysis. Duvenaud et al. (2014) visualize pathologies on the landscape of deep gaussian processes that model deep wide-limit nonlinear networks. Fourier analysis of network functions (Candès, 1999) wrt. input (Rahaman et al., 2019; Xu et al., 2019; Xu, 2018; Basri et al., 2019; Yang & Salman, 2019) has been used to show an inductive bias towards low-frequency functions (wrt. input $\mathbf{x}$), as well as a strong anisotropy of this spectrum. Wang et al. (2020) prove under some assumptions that all "bad" local minima of a deep residual network are very shallow.

## 3    HARMONIC DISTORTION

We now analyze the effect of a nonlinearity by relating the Fourier spectrum of a preactivation with that of its postactivation. Let $f$ denote the preactivation of a single neuron of a neural network consisting of $L$-layers. We use $\mathbf{x}$ to denote the input to the whole network and thus to $f$, $\mathbf{W}$ to denote

the weights, and $\phi$ to denote the employed nonlinearity. Li et al. (2018) visualize "roughness" using random 2D-slices in weight space. We follow their basic idea and consider 1D slices

$$p(t) = f(\mathbf{x}, \mathbf{W} + \alpha^{-1} \cdot t \cdot \mathbf{D}) \tag{1}$$

for random directions $\mathbf{D}$ and $t \in [0, 1]$. $\mathbf{D}$ is initialized with entries from $\mathcal{N}_{0,1}$ and normalized to $||\mathbf{D}||_F = 1$. $\alpha$ determines the path length. By varying $\mathbf{D}$, this samples a ball of radius $\alpha$ around a point $\mathbf{W}$ in parameter space. For $\phi = id$, the network $f$ is multi-linear in $\mathbf{W}$ and thus $p$ is polynomial in $t$; empirically, this yields rather smooth functions (Fig. 1). To understand general nonlinearities $\phi$ better, we represent $p$ by a complex Fourier series (Appendix B.1 discusses convergence and approximation quality):

$$p(t) = \sum_{k=-\infty}^{\infty} z_k \exp\left(2\pi i k t\right), \quad z_k \in \mathbb{C}. \tag{2}$$

Here, the sequence $z : \mathbb{Z} \to \mathbb{C}$ contains the Fourier coefficients for $p : [0, 1] \to \mathbb{R}$. As $p$ is real, $z$ is symmetric in the sense of $z_k = \overline{z_{-k}}$.

## 3.1 FORMALIZING ROUGHNESS

The roughness of a class of random functions can be characterized by the statistics of their power-spectrum (Musgrave, 1993): Given a random process that generates functions $p$, we consider mean $\mu_k$ and variance $\sigma_k^2$ of the Fourier coefficients $z_k$. For paths with short length as used in our experiments, the means $\mu_k$ are empirically very close to zero except for $z_0$ ("DC" coefficient), which is excluded in all experiments. Therefore, we can focus on variance: In general, functions where the variance of high-frequency components drop off more quickly will appear smoother. A common model, which often fits natural data well, is fractal Brownian motion (FBM-) noise, where the $\sigma_k$ drop off according to a power law:

$$\sigma_k \sim \mathcal{O}\left(1/k^h\right) \text{ for some } h > 0. \tag{3}$$

The so-called "fractal coefficient" $h$ describes the roughness of the noise function. A similar approach has been taken by Hoffer et al. (2017), who modeled the loss surface by analyzing the dynamics of a random walk on a random potential. In our experiments, we estimate the average power-spectrum $\mathbb{E}_{\mathbf{D}}(|z_k|^2)$ (by sampling $\mathbf{D}$ uniformly on a unit sphere) and fit a power-law to these spectra in order to quantify the roughness in a single number $h$.

Experiments (Section 4) and analytical arguments (Appendix B.2), show that the FBM/power-law model is a realistic model of the functions computed by the lower layers of a neural network. For higher layers the fit becomes worse, a phenomenon we will explore in the next chapter using harmonic distortion analysis.

## 3.2 WHY IS THE OUTPUT FUNCTION GETTING ROUGHER?

Intuitively, applying ReLU to a function $p$ is reminiscent of clipping an audio signal in amplitude, which is known to produce high-frequent ringing artifacts. We describe the effect of a single nonlinearity $\phi$ on the spectrum of a preactivation $p$; Inductively, this describes the spectral shift of the whole network. For the analysis, we assume that $\phi$ is a $K$-th order polynomial:

$$\phi(x) = \sum_{j=0}^{K} a_j x^j. \tag{4}$$

The effect of polynomial nonlinear maps on the spectrum of a function can be understood by harmonic distortion analysis (see e.g. Feynman et al. (1965), Ch. 50.8). We can simply (see Appendices B.1 – B.3 for details) plug the Fourier expansion of $p$ into the polynomial representation of $\phi$:

$$\phi(p(t)) = \sum_{j=0}^{K} a_j \left[\sum_{k=-\infty}^{\infty} z_k \exp\left(2\pi i k t\right)\right]^j. \tag{5}$$

The convolution theorem tells us that $j$-th power of functions corresponds to convolving the spectrum $z$ of the function $j$-times with itself. We designate by $z$ the vector containing all Fourier coefficients $z_k$ and by $\bigotimes$ the convolution operator. We can then write the spectrum of the output $z'$ as:

$$z' := \mathcal{F}(\phi(p)) = \sum_{j=0}^{K} a_j \bigotimes_{1=1}^{j} z. \tag{6}$$

**Discussion:** We can make three important observations: • First, each repeated auto-convolutions in Eq. 6 broadens the spectrum by adding higher-frequency terms. We call this broadening effect *blueshift*. The exact magnitude is hard to quantify and thus left to the experiments (Appendix B.4 gives informal arguments for a growth of $\mathcal{O}(j^{1/2})$ rather than the trivial upper bound of $\mathcal{O}(j)$). • Second and correspondingly, larger coefficients $a_j$ for larger orders $j$, increase the blueshift. • Third, $j$-fold convolutions correspond to $j$-th powers of the $z_k$. Hence, larger magnitudes $|z_k|$ also increase the blueshift (the nonlinearity becomes more visible with larger magnitude).

### 3.3 COMMON NONLINEARITIES

In practice, nonlinearities are usually not polynomial, and might not even have a globally convergent Taylor series. It is possible to approximate any continuous function by a polynomial (Stone-Weierstrass theorem). We conjecture that a close polynomial approximation will be sufficient for a qualitative prediction of the blueshift effect (our theoretical model does not guarantee convergence – we therefore validate this claim experimentally). We employ a Chebyshef approximation (which is reasonably stable and non-oscillatory) on the interval $[-5, 5]$ and compare the speed at which higher order coefficients drop off:

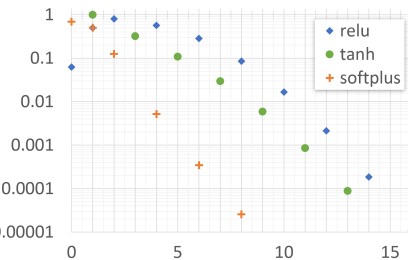

Figure 2: Absolute values of polynomial coefficients: $|t_j|$ over $j$; see also Fig. 19.

Fig. 2 shows the coefficient magnitudes for $\mathrm{ReLU}$, $\tanh$ and $\mathrm{softplus}$ (which we focus on in our experiments). $\mathrm{softplus}$ has the strongest drop-off, followed by $\tanh$ and $\mathrm{ReLU}$. Appendix F.1 gives more details and covers several additional popular nonlinearities (Figures 18, 19). Figure 3, 17 show that this correlates with blueshift, as expected.

### 3.4 CONNECTION TO EXPLODING GRADIENTS

The creation of higher-frequency harmonics directly affects gradient-based training methods because the gradient operator is a high-pass filter in the spectral domain: Taking derivatives multiplies the Fourier coefficients by $2\pi i k$, amplifying coefficients linearly with frequency $k$ (which is trivial seen by taking the derivatives of Eq. 2). The reciprocal fractal exponent $r := h^{-1}$ has exponential influence on the average gradient magnitude: With $|z_k|^2 \sim \Theta(k^{-h}) = \Theta(k^r)$, we obtain gradients $k \cdot |z_k| \in \Theta(k^{1+r/2})$. Blueshift thus causes exploding gradients: Weights of lower layers are shifted more often, creating higher-order harmonics, which correspond to larger norm of the gradient function $\|p'(t)\| = 2\pi\sqrt{\sum_k k^2|z_k|^2}$, which means that there must be, on average, larger gradients along the path considered. Independently of this, both exploding and vanishing gradients can be caused by other effects than blueshift: Saxe et al. (2014) show that even deep linear networks can suffer from exploding gradients caused by stacking matrices with non-uniform singular spectrum.

The blueshift effect itself is independent of the condition number of the weight matrices: blueshift would occur even when only stacking nonlinearities. Although it is possible to find a locally linear region of the loss surface (e.g. CReLU + Looks Linear initialization (Balduzzi et al., 2017)) and locally achieve dynamic isometry (Xiao et al., 2018), blueshift effects become relevant again when this region is eventually left during the training process (ref. Appendix Figure 12). In contrast, residual connections affect the entire loss surface and our model is able to capture this effect.

### 3.5 RESIDUAL AND MULTIPATH NETWORKS

Residual Networks exhibit *two* different dampening effects on the loss surface that can be experimentally isolated:

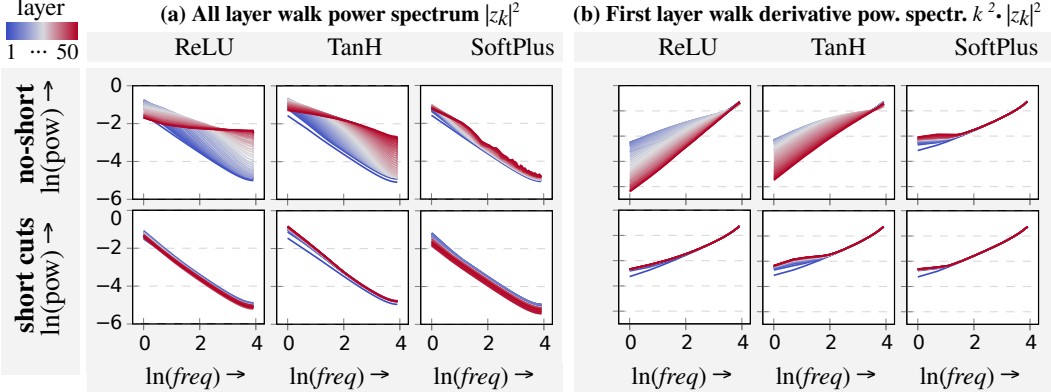

Figure 3: Normalized power spectra $z_k$ per layer (red is deeper) in a 50-layer CNN at initialization, averaged over multiple neurons, for various $\phi$. (a) On the left, all weights are varied and spectrum of $p(t)$ is shown. (b) On the right, only the first layer is varied and the spectrum of $p'(t)$ is shown.

**Exponential downweighting (ED):** Residual networks repeatedly compute the sum of the result of a small feedforward network with its input. By construction, this reduces the relative weight of the nonlinear output of the residual block by mixing it with the unprocessed signal. Doing this repeatedly creates an ensemble (Huang et al., 2016) in which the weight of contributions shrink exponentially with the number of nonlinear processing steps. As detailed e.g. in (Veit et al., 2016), each nonlinear block only contributes partially to the overall result. The weight of the signal component passing through nonlinearities shrinks exponentially due to vanishing gradient effects (at initialization, non-aligned non-uniform singular value spectra lead to exponential dampening; after training, a similar dampening effect is observed empirically by Veit et al.).

**Frequency-dependent signal-averaging (FDSA):** According to the blueshift model, the outputs of the residual blocks will contain more high-frequency Fourier coefficients (from harmonics). The additional harmonics depend on the weights in the residual block, which are statistically independent of the input (see Appendix B.5 for details). Therefore, we expect to see a reduction in the expected linear correlation between the main and residual branch for higher frequencies. As an effect, when adding the main and residual branch, higher frequencies get dampened even more.

The second effect is weaker than the first, governed by the law of large numbers (i.e., $n^{-\frac{1}{2}}$ decay for averaging $n$ independent values), but is also more broadly applicable: It should generally occur when averaging over multiple computational paths with independent weights, while residual connections require an identity in one path.

## 4 EXPERIMENTAL RESULTS: MEASURED SPECTRA

The model presented in the previous section gives us only qualitative hints on the magnitude of the blueshift, the behavior of non-polynomial nonlinearities, and the magnitude of ED and FDSA effects. We therefore now validate these qualitative predictions experimentally.

We consider two architectures: a basic Toy-CNN model with a constant number of features in each layer for fair per-layer statistics and ResNet v1 variants for a more "realistic" model. We used Cifar10 as input data, but all figures in this section look qualitatively similar on other datasets; A more detailed network description, parameters and experiments on MNIST can be found in the Appendix. In this section, we always measure the effect of blueshift in the region around initialization (i.e., $\mathbf{W}$ is the initialization point); During training, the loss surface tends to get smoother but the effect persists (ref. Appendix Figure 11).

### 4.1 EFFECTS OF BLUESHIFT ON A LOSS PATH

To show the effect of nonlinearity and residual connections on the loss surface, we sample a single loss path on a ResNet56 with Leaky ReLU activations. We can now continuously tune the amount of

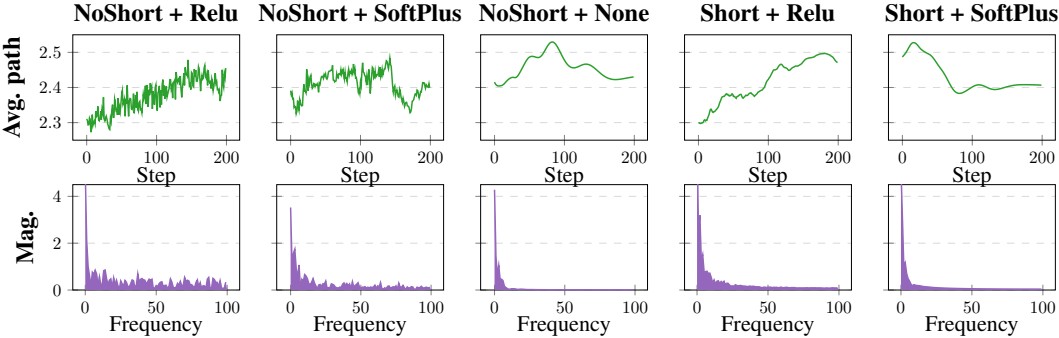

Figure 4: "Average loss path" and its respective magnitude spectrum out of 500 random loss paths for different variants of ResNet56 at initialization.

nonlinearity by adjusting its negative slope $\alpha$. We can equally continuously tune the relative weight of the residual branch by multiplying it with a factor $\nu$ in all residual blocks. Figure 1 shows a transition from a linear feedforward to a nonlinear feedforward to a nonlinear residual regime; We can observe the apparition of harmonics with increasing nonlinearity that get dampened when switching on residual connections.

## 4.2 Effects of blueshift on the average power spectrum of the w-o surface

We now want to view the effects of blueshift on the expected power-spectrum of the w-o surface. We use a "Toy-CNN" with depth $L = 50$ and track the w-o surfaces of each layer while we walk in a single random direction in weight space. We transform the resulting 1D-paths with discrete FFT and average all power spectra over all neurons in a layer, all input receptive fields and all batch images. Fig. 3a (left side) shows the resulting average power spectrum, normalized by function norm. The right hand side (b) shows the same plot, but with $\mathbf{D}$ restricted to only changing the weights of the first layer of the network, and taking the derivative of the resulting function (by weighting by frequency, $k^2 \cdot |z_k|^2$) before normalization. As we use only 100 samples without prior bandlimiting, aliasing phenomena occur as a slight upward slant is visible at the high-frequency end of all plots.

**Spectral shift over depth:** The results in Fig. 3a confirm our predictions quite well: A spectral blueshift is clearly visible and, as expected, $\mathrm{ReLU}$ shows a strong blueshift, $\tanh$ a slightly weaker blueshift, and $\mathrm{softplus}$ only a small effect. Harmonics show up as a bump in the spectrum that travels towards higher frequencies with increasing layer. This extends to other nonlinearties as well (see Appendix, Fig. 17). The plot also confirms that the power spectrum of the lower layers is well described by a $1/k$-power law, as predicted.

**Scale shift of a single layer parametrization:** By only changing the weights in a single layer and taking the derivative (Fig. 3b), we see that blueshift is responsible for a gradient scale mismatch between earlier and deeper layers in networks without skip-connections: the more nonlinearities are between the modified weight and the output, the higher the gradient magnitude.

**Residual connections:** Our model predicts that residual connections will reduce the blueshift strongly by **ED** and **FDSA** (ref. Section 3.5). This is also consistent with our observations: higher layers show almost the same $1/k$-spectrum as the initial layer. Looking at the first layer weights, weighted by frequency, the response is almost flat, with a small emphasis on low-frequency weights.

## 4.3 Quantifying and measuring spectral shift

Utilizing the power-law model presented in section 3.1, we can measure blueshift by estimating the fractal exponent $h$ of a loss path via a power-law fit of the magnitude spectrum. Figure 4 represents what the "average loss path" looks like for a ResNet 56 at initialization with different activation functions, with and without shortcuts. For each architecture, we sample paths in 500 different random directions $\mathbf{D}$ and assess the fractal exponent of every sampled path, in order to visualize the path

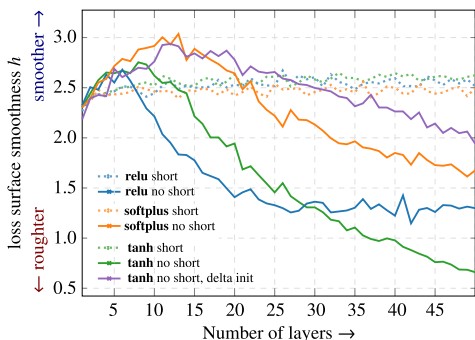

Figure 5: Smoothness of the loss surface at initialization for a Toy-CNN of varying depth with and without shortcuts.

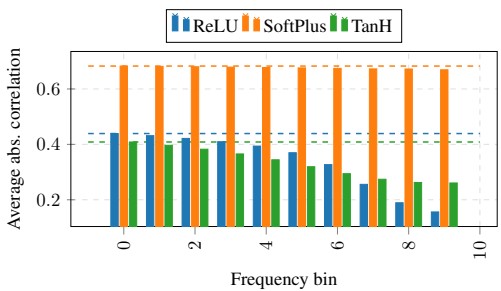

Figure 6: Average absolute path correlation per frequency bin of the main branch and the residual branch of a ResNet56 v2 with different activation functions at initialization. Lower bin number indicates lower frequencies.

with median $h$ value and its associated spectrum. We clearly see that nonlinear architectures without skip-connections present most blueshift, the linear architecture has no blueshift and the architectures with skip-connections do also exhibit, albeit dampened, blueshift.

In Figure 5, we assess the average loss path smoothness $h$ of a Toy-CNN with varying depth at initialization. For averaging, we initialize each network 50 times and sample 20 paths in different random directions for each initialization. The results confirm quantitatively the visual results of Li et al. (2018) that the loss surface becomes rougher with increasing network depth and that shortcut connections have a strong smoothing effect. Since our roughness measure is a non-pointwise view, we see that even the delta-initialized network that does not suffer from vanishing gradients at initialization still gets rougher with increasing depth. The power-law fit still has some limitations: As seen in Fig. 3, a power-law fit is imperfect as the higher layers are not straight lines anymore. This shows in curves of Fig. 5 as apparent increase of smoothness in the first layers for the "NoShort" architectures.

## 4.4 RESIDUAL NETWORKS CONTROL HARMONICS VIA ED AND FDSA

In Section 3.5, we described two different mechanisms that allow ResNets to dampen high-frequency modes. We now want to verify the existence of **FDSA** and further individuate the two effects.

**FDSA** predicts that skip-connections have a dampening effect on high frequencies in the loss surface because the correlation between the main branch and the residual branch is decreasing with higher frequencies. We empirically verify this prediction on a ResNet 56 v2 at initialization by walking in 50 random directions and measuring the respective w-o function (we only sample 10% of the outputs for computational reasons) of the two summands in each residual block just before addition. Using Gaussian filters in Fourier domain, we can compute the linear correlation of each pair of paths (block input/output) per frequency bin and average over all batch images, neurons and blocks in the network. In Figure 6, we see that the absolute linear correlation is decaying for all architectures with increasing frequency, with $\mathrm{ReLU}$ showing the strongest effect, again followed by $\tanh$ and $\mathrm{softplus}$ in that order, confirming our prediction. This property seems to allow a residual network to automatically regulate the higher-frequency harmonics content of the output.

We want to individuate **FDSA** in a setting where **ED** is not present. We modify our Toy-CNN and ResNet "NoShort" architectures such that we replace every $n$-feature *convolution / batch normalization / activation* sequence in the network with a $a \cdot n$ feature sequence where the $a$ feature groups get averaged after the activation (in reality we only use summation, since normalizing is taken care of by batch normalization). Since high frequencies added by the nonlinearity are less correlated between features (ref. Fig. 6), we expect that averaging feature groups smoothen the w-o function; we see on the right of Figure 8 that this is indeed the case.

Finally, we want to demonstrate that the positioning of the activation function is crucial. Our model predicts that since ResNet v2 blocks (He et al., 2016b) use summation after ReLU, we will see a stronger averaging effect than in ResNet v1. Indeed, looking at their respective average power spectra in a random direction in Figure 7, we see this confirmed.

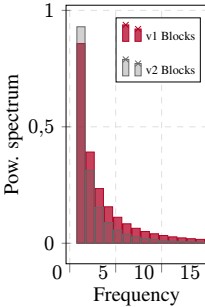

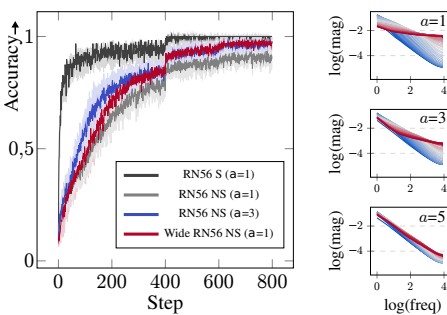

Figure 7: Normalized power spectrum of the last layer of a ResNet194 v1/v2 (post/pre-activation) at initialization.

Figure 8: **Left**: Comparing the training performance on Cifar10 of a feature-averaging ResNet56 v1 "NoShort" to its vanilla, wide and residual equivalent. **Right**: Spectrum shift of averaging Toy-CNN 50 layers at initialization, varying averaging factor $a$.

## 5 EMPIRICAL RESULTS: TRAINING

In this section, we want to demonstrate a correlation between blueshift and network trainability. Again, we use variants of the ResNet v1/v2 network, exact network parameters and training hyperparameters can be found in tabular form in Appendix A. We use Cifar10 as training data, training results for Cifar100 can be found in Appendix C

**Controlling Blueshift via feature averaging:** We want to investigate the impact of the smoothing of the loss surface via **FDSA** (ref. Section 3.5) on training speeds. We choose a ResNet56 "NoShort" as our architecture, since at this depth simple feedforward networks with batch-normalization start to become difficult to train. We experiment on the averaging network (ref. Section 4.4) with $a = 3$ since we experimentally determined that for $a > 3$ the performance stops increasing. For fairness, we included a Wide ResNet56 "NoShort" with approximately the same parameter count than the averaging network. We see the averaged results over five runs in Figure 8 (shaded areas represent the standard deviation). We observe that the averaged network outperforms both the original network and the wide network while still performing worse than the network with skip-connections. This is consistent with our theory since **ED** has a bigger smoothing effect than **FDSA**.

**Controlling Blueshift with Leaky ReLU:** As demonstrated in Figure 1, the amount of blueshift in a network can be controlled by tuning Leaky ReLU's negative slope $\alpha$. We now want to show that networks with a very strong blueshift are hard to train but conversely some amount of nonlinearity is needed for expressivity. For this, we train a ResNet56 v1/v2 with and without shortcuts for 30 epochs once on Cifar10 for different values of $\alpha$ and visualize the training accuracy (average over the last 25 values) and compare it to the blueshift of the network at initialization. On Figure 9, we see that for the networks without skip-connections, training becomes more difficult with regard to network depth, which correlates with increased blueshift. We see that this effect can be alleviated by making the network more linear (by increasing $\alpha$) and thus reducing blueshift. For networks with skip-connections, we see that blueshift is greatly reduced and that no deterioration of trainability with regard to network depth is noticeable. Conversely, networks that near a linear regime for high values of $\alpha$ tend to train worse since they lack expressivity. Interestingly, this effect is stronger for ResNet v2 blocks than for ResNet v1 blocks because the former make the loss surface smoother than v1 blocks (ref. Figure 7).

## 6 DISCUSSION

Our experimental findings validate the predictions of our model with respect to the roughness of the w-o function: Nonlinearities with larger higher-order polynomial coeffiecnts create larger variances in the high-frequency part of the spectrum. While related results on increasing complexity with depth have been given earlier (Poole et al., 2016; Schoenholz et al., 2017), the harmonics model offers a simple and more global view of the roughness of the loss surface and how it is affected by nonlinearity choice and residual connections. Our observations on training speed are consistent with the hypothesis that spectral blueshift impedes training: Architectural choices (residual connections,

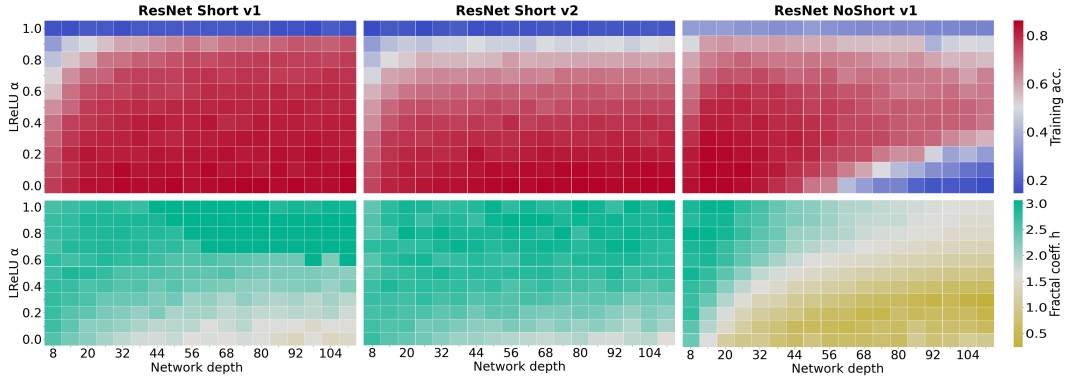

Figure 9: Comparing w-o smoothness and trainability for ResNets with varying depth and amount of nonlinearity. **Upper row**: training accuracy after 30 epochs of training on Cifar10. **Bottom row:** blueshift at initialization indicated by the fractal coefficient of the network's last layer.

ResNet v2 blocks, LReLU) that control harmonics generation are also those that are easier to train to the their full performance. Our model is also consistent with the empirical observations of Han et al. (2017), who found that ResNet performance varies with the location and number of ReLUs. Further, the results in Figure 9 give strong hints for a correlation between high-frequency content and problems in trainability for deep but very nonlinear networks (small $\alpha$). From an analytical perspective, architectures that are prone to blueshift are prone to vanishing/exploding gradients: We have shown that high magnitudes of high-frequencies in the w-o functions lead to an increase in the expected gradient magnitude. Further, in a simple multi-layer network without shortcuts, we expect more high-frequency content on lower than on higher layers as soon as the nonlinearity of the activation function is actually is utilized (and therefore must also creates harmonics). A rescaling of layers can reduce these discrepancies, but the normalization will still necessarily depend on the position in parameter space. As far as accuracy is concerned, there is an obvious trade-off between blueshift and expressivity: networks with low-degree polynomials as nonlinearities contain less high-frequent harmonics but cannot approximate complex functions (Kidger & Lyons, 2020). Future work could consist in further exploring this trade-off. Further, it would be interesting to examine potential architectural alternatives to residual connections with comparable performance and trainability.

## 7    CONCLUSION AND FUTURE WORK

In this paper, we have analyzed the effect of different architectures of nonlinear networks on the smoothness of the computed function. Specifically, we have linked polynomial nonlinearity to harmonics creation and validated experimentally that the mean power spectrum shifts towards higher frequencies ("blueshift"). Further, we have described two distinct effects that explain the smoothing effect of shortcut connections, which we also confirmed experimentally. Finally, we empirically linked reduced blueshift to increased trainability and training speeds.

In future work, we hope to derive explicit closed-form equations for how architectural designs like network depth, residual connections and nonlinearity choice affect the w-o function. It could be also interesting to further explore the roughness-expressability trade-off that we described earlier.

## 8    ACKNOWLEDGEMENTS

This work has been partially supported by the RMU Network for Deep Continuous-Discrete Machine Learning (DeCoDeML project). We wish to thank Jan Disselhoff for valuable discussions and the annonymous reviewers for their helpful feedback.

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

## A  NETWORK DETAILS

### A.1  NETWORK ARCHITECTURES

The implementation for our experiments is based on PyTorch 1.5 and are provided as supplementary material.

**Toy-CNN:** We create a simple convolutional network consisting of units of 3x3-convolutions with 16 feature channels on all layers ($16 \cdot a$ for the averaging networks), followed by batch normalization and non-linear activation, stacked-up $L$-times. Optional short-cut connections are added (only if explicitly stated) between each unit (i.e., residual blocks of depth 1). No pooling or striding is used. The input layer uses a stride of 5 to reduce memory demands. The classification layer is composed of a fully-connected layer, softmax and cross-entropy loss.

**Residual Network:** As a "real-world" example, we use a standard Residual Network v1 (BasicBlock) He et al. (2016a) with varying depth, with ("Short/S") and without ("NoShort/NS") skip-connections. ResNet v2 blocks are used in Figure 1 (v2 blocks show slower transitions) and 6 (correlations are decaying faster in ResNet v2, ref. 3.5). We use the standard number of planes for Cifar-10 training : 16/32/64 (28/56/112 for the wide network).

## A.2  FOURIER WALK HYPERPARMETERS

We sample paths consisting of 100 samples each and with path length $\alpha = 1$ (Eq. 1) in all experiments, except for Figures 1, 4 and 6 where we double both values for higher frequency resolution and Figure 12, where we want to walk in a different radius ($\alpha \in \{0.1, 1, 10\}$). All experiments are performed in training mode and with data augmentation at initialization. Datapoints for each architecture are measured on a different mini-batch to exclude bias by batch selection.

## A.3  TRAINING HYPERPARMETERS

The hyper-parameters below usually reach the standard test-accuracy of approximately 92-93% for a ResNet56 on Cifar10.

| | |
|---|---|
| Dataset | Cifar10 (Cifar100 for Figure 10) |
| Epochs | 200 |
| Scheduler | Multistep ($\gamma = 0.1$) |
| Milestones | 100, 150 |
| Learning rate | 0.1 |
| Batch size | 128 |
| Optimizer | SGD + Momentum |
| Momentum | 0.9 |
| Weight decay | 0.0001 |
| Augmentation | Random Flip |

## B  FORMAL DETAILS CONCERNING HARMONIC NETWORK ANALYSIS

### B.1  SERIES REPRESENTATION AND CONVERGENCE

In our paper, we consider feed-forward networks consisting of $L$ layers. Each layer $l = 1...L$ can be one of the following:

- An **affine map**
$$\mathbf{x} \mapsto \mathbf{W}^{(l)}\mathbf{x} + \mathbf{b}^{(l)} \tag{7}$$
  that transforms the input using a linear map and an offset vector.
- A **nonlinearity**
$$\mathbf{x} \mapsto \phi^{(l)}(\mathbf{x}) \tag{8}$$
  that applies a (potentially layer-specific) nonlinear function to each input.

All of the discussed architectures (including convolutional networks, striding/average pooling, residual connections, parallel computations, and layer-dependent nonlinearitiy) can be expressed as a composition of these two layer types. We currently do not address max-pooling, and understand batch-normalization layers statically as a fixed scaling and shifting (not analyzing their training dynamics). The loss function itself (such as a sequence of softmax and cross-entropy) is also not included in our analysis – we restrict our consideration to layer outputs before any classification. As logistic regression (or similar problems) are numerically well-understood already, this is not a major obstacle. All our experiments show layer outputs before classification (softmax).

The main restriction of our formal model is that we assume that all nonlinearities are polynomials of finite degree[1]. Under this condition, all outputs and intermediate results of the whole network can be

---

[1]This means that the analytical results are currently not established for practical non-polynomial nonlinearities. The results in our paper obtained from polynomial approximations should therefore be considered experimental results at this point. An analysis of infinite series approximations covering a more general class of nonlinearities is still left for future work.

represented by a large multi-variate polynomial, and any *linear section* $p(t) = f_i^{(l)}(\mathbf{W} + \alpha t\mathbf{D}, \mathbf{x})$ of any output $i$ of any layer $l$ is a univariate polynomial of finite degree.

**Fourier analysis:** Polynomial functions on a finite domain are Lipschitz-continuous and have finite variation; both conditions are sufficient for the convergence of the Fourier series in an $L_2$-sense. Further, as the Fourier-basis forms a Schauder-basis of $L_2[0, 1]$, the series expansion is unique.

In our derivation (Equation 5), we plug the series representation into the polynomial and conclude that it can be represented in the spectral domain as a sum of convolutions (Equation 6). To make this more rigorous, we proceed in smaller steps: Let again $p : \mathbb{R}^n \to \mathbb{R}$ denote the linear section along a linear direction in parameter space of any layer output. Our nonlinearity expands to

$$\phi(p(t)) = \sum_{j=0}^{K} a_j p(t)^j. \tag{9}$$

As $p(t)$ is a polynomial of finite degree, $p(t)^j$ for any finite $j$ is also a polynomial of finite degree. This implies that there is a unique Fourier series that represents $p$, characterized by the coefficients sequence $z : \mathbb{Z} \to \mathbb{C}$. According to the convolution theorem for the complex Fourier series, the Fourier series of $p \cdot p$ is given by $z \otimes z$, where

$$[z \otimes u]_k := \sum_{m=-\infty}^{\infty} z_m \cdot u_{k-m}. \tag{10}$$

Because the series must exist (it converges for both $p$ and $p^2$) and it is unique, the product of the series must be represented by the auto-convolution. Inductively, we obtain $p^j$ corresponding to the $j$-fold auto-convolution $\bigotimes_{q=1}^{j} z$.

**Note:** We use the notation $[\mathbf{x}]_i := x_i$ denote the indexing of vectors and sequences that result from a computation.

**Approximation quality and finite transforms:** The Fourier representation is accurate in an $L_2$-sense; it converges in $L_2$-norm, which is an integral measure. This still permits any finite series expansion to have large point-wise errors (no convergence in infinity-norm: large error regions rather shrink to a zero set in the limit). A common example of such problems is Gibb's phenomenon: A discontinuous function (such as the gradient function of a ReLU) will show "overshooting" of constant magnitude near the discontinuities; only the area affected shrink with the frequency order of the Fourier series. Further, even higher-order discontinuities lead to significant high-frequency harmonics (i.e., show bad approximation behavior for truncated series).

While the theory is not affected by this, this might appear to be a concern for the discrete Fourier transformation (DFT) we use in our experiments. The DFT operates at a finite frequency order, using a matching set of regular samples of the continuous function. Here, it is important to note that the DFT, as a mapping from $\mathbb{C}^d$ to $\mathbb{C}^d$ will reproduce the function values faithfully at each sample point for any frequency order (the DFT is bijective and even unitary, thus even allowing numerically accurate and stable reconstructions). Evaluating the obtained series in between sampling points, however, would reveal issues at discontinuities; as in most applications of DFT, our experiments do not perform such continuous evaluations.

An issues that does remain is aliasing: Nonlinearities in general and in particular discontinuous ones (even with higher order discontinuities) broaden the spectrum (this effect is particularly strong for first-order discontinuous functions such as ReLU). Therefore, the sampling frequency has to be chosen sufficiently high to capture these effects. Otherwise, the high-frequency harmonics will alias as lower-frequency signal components, which might lead to an underestimation of the blueshift effect.

### B.2 INPUTS TO THE SECOND LAYER NONLINEARITIES ARE FBM-NOISE

For the first layer of a ReLU network, the following calculation also supports the experimental findings of an $\mathcal{O}(1/k)$-power law from an analytical perspective. Let $p^{(1)}$ be a w-o path of the output of a given neuron from the first layer. It follows that:

$$p_i^{(1)}(t) = \text{ReLU}\left[\sum_j \left(w_{ij}^{(1)} + \alpha^{-1}t \cdot d_{ij}^{(1)}\right) \cdot x_j\right] \tag{11}$$

$$= \text{ReLU}\left[m \cdot t + n\right], \text{ for some } m, n \in \mathbb{R}. \tag{12}$$

This is a piecewise linear function that is constant zero on one interval and linear with slope $m$ on another interval. With $a < b \in [0, 1]$ bounding the activation interval, the Fourier series becomes:

$$z_k^{(1,i)} = \int_a^b \exp\left(-2\pi i k \left(mt + n\right)\right) dt. \tag{13}$$

By the chain rule of derivations, this implies

$$|z_k^{(1,i)}| \in \mathcal{O}\left(\frac{1}{k}\right). \tag{14}$$

Correspondingly, the preactivations of the second layer will be random linear combinations of FBM-noise with $h = 1$, which again forms FBM-noise with that spectral variance (every Fourier coefficient is a sum of independent coefficients; therefore, variance is additive).

Similar findings where also previously shown in input space by Balduzzi et al. (2017), which, for the first layer, is just the dual to varying the weights (and therefore must yield the same result).

### B.3 BLUESHIFT AND DEPTH

We now aim at formally understanding the blueshift effect in a multi-layer network. For simplicity, we consider a simple stack of linear and nonlinear layers with a fixed nonlinearity:

$$f(\mathbf{x}, \mathbf{W}) = \phi\left(\mathbf{W}^{(L)}\phi\left(\mathbf{W}^{(L-1)}\cdots\phi(\mathbf{W}^{(1)}\mathbf{x})\cdots\right)\right). \tag{15}$$

We now consider a single output $i$ of layer $l$:

$$f_i^{(l)}(\mathbf{x}, \mathbf{W}) = \phi\left(\mathbf{w}_i^{(l)} \cdot \phi\left(\mathbf{W}^{(l-1)}\phi\left(\mathbf{W}^{(l-2)}\cdots\phi(\mathbf{W}^{(1)}\mathbf{x})\cdots\right)\right)\right). \tag{16}$$

Next, we consider a linear section $t \mapsto \mathbf{W} + t\mathbf{D}$ with $t \in [0, \alpha]$:

$$p_i^{(l)}(t) = \phi\left(\mathbf{w}_i^{(l)} \cdot \phi\left((\mathbf{W}^{(l)} + t\mathbf{D}_l)\phi\left((\mathbf{W}^{(l-1)} + t\mathbf{D}^{(l-1)})\cdots\phi(\mathbf{W}^{(1)} + t\mathbf{D}^{(1)}\mathbf{x})\cdots\right)\right)\right).$$

By replacing $\phi$ with finite polynomials (Equation 4), we obtain multi-variate polynomials as outputs of all of these function.

**Varying parameters in a single layer:** If we now select a layer $k$ by setting all $\mathbf{D}^{(l)} = \mathbf{0}$ for all $i \neq k$, and only using the $k$-th matrix as direction vector, the parameter $t$ will pass through $l - k + 1$ nonlinearities:

$$p_i^{(l)}(t) = \phi\left(\mathbf{w}_i^{(l)} \cdot \phi\left(\mathbf{W}^{(l-1)}\cdots\phi\left((\mathbf{W}^{(k)} + t\mathbf{D}^{(k)})q^{(k-1)}\cdots\right)\cdots\right)\right). \tag{17}$$

with a constant vector $q^{(k-1)}$.

Correspondingly, each nonlinearity will act on the spectrum of $q^{(k)}$ as a blueshift operator (forming the sum of repeated auto-convolutions of the spectrum, weighted by the polynomial coefficients). Therefore, we see that parameters of earlier layers (small layer index $k$, closer to the input layer) are blueshifted more frequently than later layers.

The equation also shows that the singular value spectrum of $W^{(l)}$ affects the results as well, in addition to the spectral shifts caused by the nonlinearities: the values rescale the input domain

(shrink/expand space wrt. parameter $t$), thereby changing the frequency reciprocal to the scale factor. However, a linear transformation scales the Fourier spectrum as a whole while nonlinearities spread the spectrum irreversibly by applying blueshifts (weighted sums of auto-convolutions of the spectrum). The magnitude of this nonlinear effect is, nontheless, dependent on the signal magnitude and thus affected by choices of $\mathbf{W}^{(l)}$.

**Varying parameters in layer** $0..l$**:** Similar arguments hold if we vary only parameters in the first $l$ layers of the network. However, we obtain a mix of spectra that have been blueshifted (and scaled by weight matrices) a different number of times.

### B.4    Spectral Broadening through Auto-Convolutions

The exact amount of broadening of a spectrum $z : \mathbb{Z} \to \mathbb{R}$ by an auto-convolution is non-trivial to quantify; we therefore resort to experiments for determining the effect in practice.

In special cases, we can analyze the effect:

**Band limited functions:** Assuming that $z$ is bandlimited, specifically $z_k = z_{-k} = 0$ for all $k > K$, and $|z_k| = |z_{-k}| > 0$ for $k \leq K$ it is trivial to see that an auto-convolution extends the support by $k$ entries to each side ($2k$ overall). This means, after $j$ convolutions, we obtain a non-zero spectrum within $k = -jK...jK$. The proof is obtained by considering the left-most and right-most terms in the definition of the discrete convolution.

**Central limits:** If we assume that $z$ is real and positive, and has finite second moments, the central limit theorem guarantees convergence towards a Gaussian function with standard deviation $\mathcal{O}\left(\sqrt{j}\right)$. The first assumption is obviously unrealistic in practice.

In the general case the $j$-th power of a Fourier series

$$\left[\sum_{k=-K}^{K} z_k \exp\left(2\pi i k t\right)\right]^j \tag{18}$$

expands to the sum

$$\sum_{k_1=-K}^{K} \cdots \sum_{k_j=-K}^{K} z_{k_1} \cdots z_{k_j} \exp\left(2\pi i (k_1 + k_2 + \cdots + k_j)\right). \tag{19}$$

If we only count the number of terms of the same frequency $k = k_1 + \cdots + k_j$, the number of occurrences of each frequency $k$ will tend towards a normal distribution with variance $j$. However, as the complex coefficients can cancel out, the limit distribution is only an upper bound.

### B.5    Frequency Dependence in Averaging of Multiple Computation Paths

We observe that averaging functions after nonlinearities leads to some smoothing of the result by decreasing the correlation between higher-frequency Fourier coefficients. In order to understand the effect, we look at a simple 1D model case: We consider a the composite functions

$$f_{post}(x) = \phi\big(\underbrace{w_1 \cdot g(x)}_{=:h_1'} + \underbrace{w_2 \cdot g(x)}_{=:h_2'}\big) \tag{20}$$

and

$$f_{pre}(x) = \underbrace{\phi(w_1 \cdot g(x))}_{=:h_1} + \underbrace{\phi(w_2 \cdot g(x))}_{=:h_2} \tag{21}$$

where $w_1, w_2$ are random numbers, drawn from the same normal distribution.

Let $z_k$ be the sequence of Fourier coefficients of $g$, and $u_k$ that of the result. In the first case, $f_{post}$, a random linear combintation of the input spectra is performed, which is then blueshifted:

$$u = \sum_{j=1}^{K} \bigotimes_{q=1}^{j} \left(w_1 z + w_2 z\right) \tag{22}$$

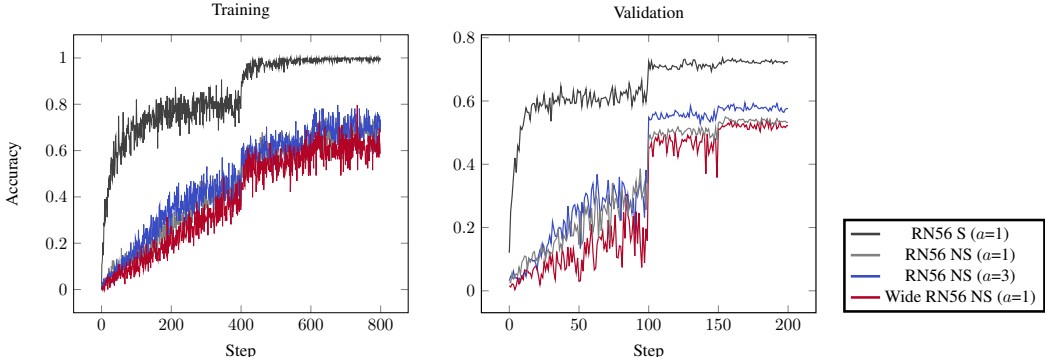

Figure 10: Repeating the experiments of Figure 8 for a single run on the Cifar100 dataset.

In the second case,

$$u = \sum_{j=1}^{K} \bigotimes_{q=1}^{j} (w_1 z) + \sum_{j=1}^{K} \bigotimes_{q=1}^{j} (w_2 z), \tag{23}$$

we first blueshift the randomly-scale spectra and then form the sum. This creates a frequency dependence because larger powers (larger values of $j$, corresponding to more repeated convolutions, and thus $j$-fold products of input Fourier coefficients) behave more nonlinearly and thus react more strongly to the weight scaling: While a linearly weighted average will be linearly correlated with the original, averages of nonlinearly transformed functions will loose linear correlation (and thus appear more random to a simple averaging operation). The harmonic distortion analysis that expresses the powers as auto-convolutions of the spectrum shows that higher-frequency components (created by the blueshift) are also the ones that behave more nonlinearly.

## C   TRAINING ON CIFAR100

We repeat the experiment on averaging-networks for the Cifar100 dataset, holding out 1% of the training data for validation. We leave all hyperparameters untouched. In Figure 10, we still see the advantage of the averaging-network over the regular network, especially in validation accuracy.

## D   OTHER SPECTRAL SHIFT FIGURES

### D.1   SPECTRAL SHIFT DURING TRAINING

To show that the blueshift effect is not limited to the initialization, we show the spectral shift of a Toy-CNN 50 at 0 and 20 epochs of training in Figure 11. ResNets still do not show any major blueshift at any time. The ReLU and TanH architectures without skip-connections show less show blueshift after 20 epochs of training, indicating that blueshift is more of a problem in early phases of training (Frankle et al., 2020). When increasing the sampling radius to $\alpha = 10$, blueshift is clearly visible again for ReLU and TanH activations in the networks without skip-connections.

### D.2   SPECTRAL SHIFT WITH DELTA-ORTHOGONAL INITIALIZATION

The experiments of Figure 3 with batch normalization are subject to exploding gradients. Xiao et al. (2018) describe an initialization scheme that fixes exploding gradients for TanH activations. Fig 12 shows that for a very small region near the orthogonal initialization point, blueshift vanishes. As soon as we walk further from the initialization point, blueshift becomes visible again.

### D.3   SPECTRAL SHIFT IN GRADIENT DIRECTION

The subspace of the loss surface where gradient descent operates has been studied (Gur-Ari et al., 2018) and attributed a specific behavior. We want to see how blueshift behaves when slicing the

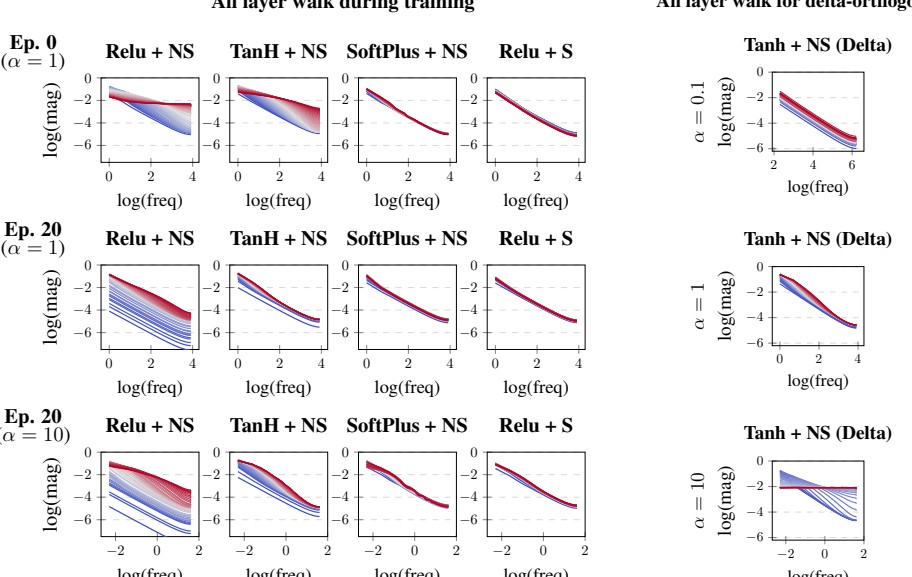

Figure 11: Repeating the experiments of Figure 3 (all layers, unscaled) during training.

Figure 12: Repeating the experiments of Figure 3 (all layers, unscaled) for TanH activations with delta-orthogonal initialization for different path lengths $\alpha$.

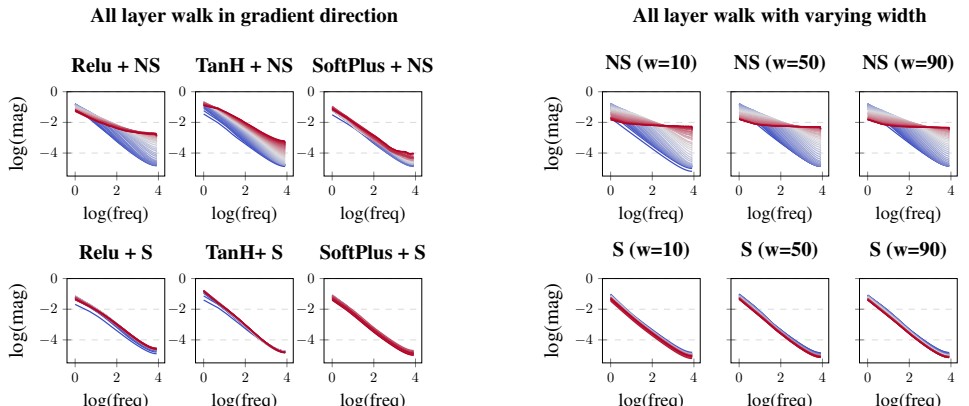

Figure 13: Repeating the experiments of Figure 3 (all layers, unscaled) at initialization in gradient direction.

Figure 14: Repeating the experiments of Figure 3 (all layers, unscaled) for variable network width.

network in the direction of gradient descent. For this, we repeat the experiments of Figure 3, but compute the gradient at initialization, normalize it and use it as direction $\mathbf{D}$ of equation 1. In Figure 13, we see that the effect is still visible although dampened for ReLU and tanh activations. The effect seems a little more visible for softplus activations.

## D.4 VARYING WIDTH

To see if layer width has any influence on blueshift, we repeat the experiment of Figure 3 for a Toy-CNN 50 with number of filters $w$. We can see on Figure 14 that layer width doesn't seem to have a notable impact on blueshift strength.

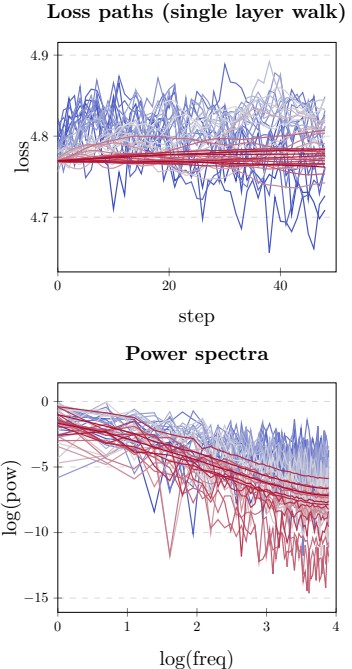

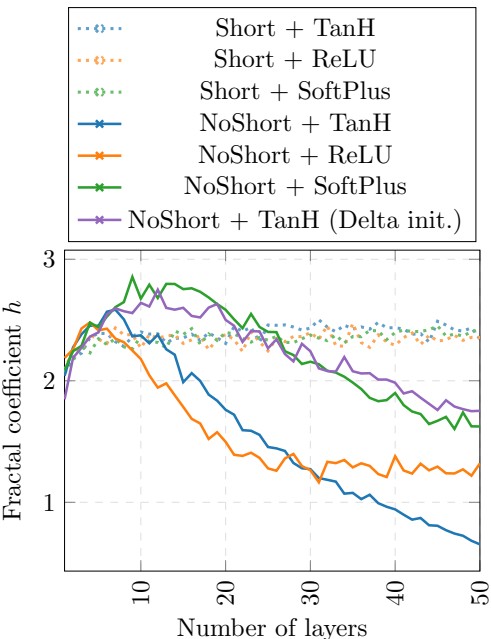

Figure 15: Resulting loss paths and respective path-normalized power spectra when varying parameters in only one layer of a Toy-CNN 50 at initialization. The color of a path indicates in which layer parameters were varied (red is deeper).

Figure 16: Smoothness of the loss surface at initialization for a Toy-CNN of varying depth with and without shortcuts on MNIST for a batch size of 256.

# E    WALK PER LAYER

To see the contribution of each layer to a loss path, we sample a random direction $\mathbf{D}$ and measure the resulting loss path when only modifying weights in a single layer. We realize this for a Toy-CNN with 50 layers at initialization in Figure 15. We see that varying weights in higher layers (red) results in smooth paths whereas varying weights in lower layers (blue) results in rougher paths. In the normalized power-spectra, we can see the increased blueshift of paths resulting from varying earlier layers.

# F    SMOOTHNESS MEASUREMENTS ON MNIST

To show that our smoothness measurements are mostly independent of batch size and dataset, we revisit Figure 5. This time, we use the MNIST dataset and a batch size of 256. All other hyperparameters are maintained. We see that although the absolute values slightly differ, the relative behavior of the nonlinearities is qualitatively the same. The results are shown in Figure 16

## F.1    MORE ACTIVATION FUNCTIONS

To demonstrate that blueshift occurs with every nonlinear activation function, we repeat the experiment of Figure 3 for exponential linear units (ELU), gaussian error linear units (GELU), hard tanh (HTANH), leaky relu (LReLU), scaled exponential linear unit (SELU) and sigmoid activations.

In order to demonstrate the that spectral spread depends on the magnitude of higher-order polynomial coefficients, we show the plot of the magnitude of the coefficients of a polynomial approximation with Chebyshev nodes of degree 25 in Figure 19. The fit is performed for the interval $[-5, 5]$ (slightly different from Figure 2 in the main paper, which shows a fit for [-1,1]). Figure 19(a) also shows again

**All layer walk with more activation functions**

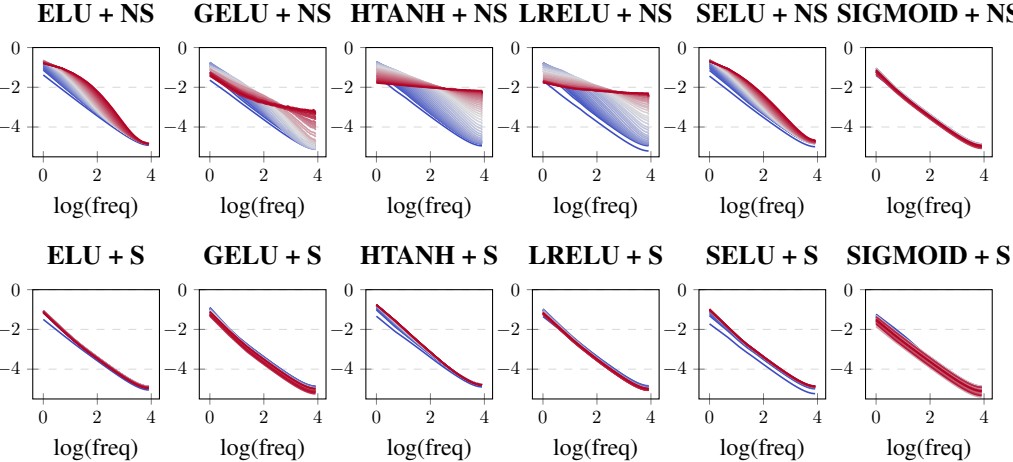

Figure 17: Repeating the experiments of Figure 3 (all layers, unscaled) for more nonlinearities.

**Polynomial Chebyshev approximation of various nonlinearities of degree 25 within [-5,5]**

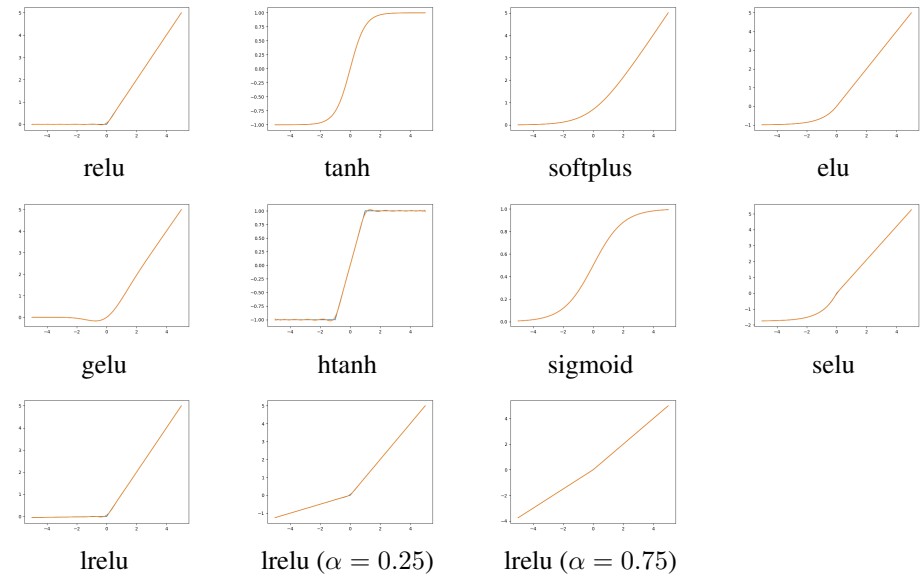

Figure 18: Various Nonlinearities with their Chebyshev-Polynomial Approximation within the interval $[-5, 5]$.

the comparison between ReLU, tanh, and softplus, uniformely with this Chebyshef approximation; this shows the differences more clearly (please also note the logarithmic scale of the $y$-Axis).

The approximations are quite tight within the interval, see Figure 18 for reference.

**Comparison of the magnitude of the polynomial coefficients $|a_j|$**
(for the degree 25 approximation of Figure 18)

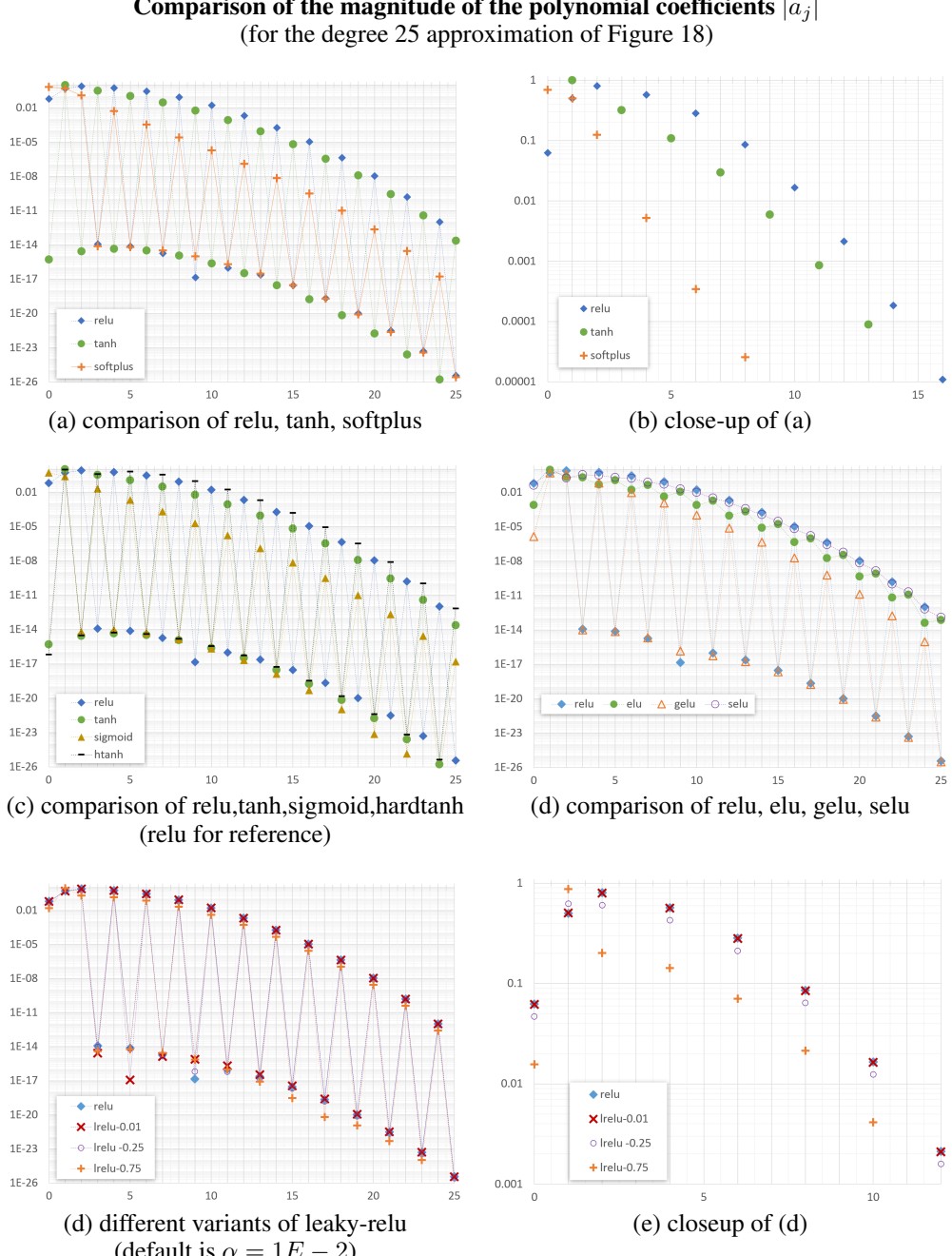

Figure 19: Absolute value of the polynomial coeffients of the Chebyshev polynomial approximation. All fits are done within the interval $[-5, 5]$. Note: Different interval from Fig. 2 in the main paper! (to capture the shape of all nonlinearities well).

| | 1 | $t$ | $t^2$ | $t^3$ | $t^4$ | $t^5$ | $t^6$ |
|---|---|---|---|---|---|---|---|
| relu | 6.2512213E-02 | 5.0000000E-01 | 8.0859691E-01 | 1.1924015E-14 | -5.7216603E-01 | -7.6820090E-15 | 2.8328407E-01 |
| tanh | 5.1805002E-16 | 9.9793926E-01 | -2.7609686E-15 | -3.2070438E-01 | 4.6215225E-15 | 1.0881659E-01 | -3.2234542E-15 |
| softplus | 6.9314722E-01 | 5.0000000E-01 | 1.2499948E-01 | -8.0003321E-15 | -5.2070252E-03 | 6.6261234E-15 | 3.4591520E-04 |
| elu | 7.9148528E-04 | 9.4044077E-01 | 2.1895988E-01 | -2.0584738E-01 | -4.8542686E-02 | 1.1656771E-01 | 1.6778199E-02 |
| gelu | 1.4022057E-06 | 5.0000000E-01 | 3.9892182E-01 | 9.9860124E-15 | -6.6440646E-02 | -7.1476355E-15 | 9.9256757E-03 |
| htanh | -6.4522701E-17 | 9.5248313E-01 | 3.1201720E-15 | 3.9176956E-01 | -5.5233592E-15 | -6.5558052E-01 | 4.0680108E-15 |
| lrelu | 6.1887091E-02 | 5.0500000E-01 | 8.0051094E-01 | 2.6432027E-15 | -5.6644437E-01 | 1.2012168E-17 | 2.8045123E-01 |
| selu | -4.2829527E-02 | 1.2996891E+00 | -1.8704691E-01 | -3.6190014E-01 | 3.1940644E-01 | 2.0493761E-01 | -1.7089694E-01 |
| sigmoid | 5.0000000E-01 | 2.4999950E-01 | -6.5774789E-15 | -2.0830617E-02 | 9.6140255E-15 | 2.0789179E-03 | -5.0960174E-15 |
| lrelu -0.25 | 4.6884160E-02 | 6.2500000E-01 | 6.0644768E-01 | 9.4213822E-15 | -4.2912452E-01 | -6.5801233E-15 | 2.1246305E-01 |
| lrelu -0.75 | 1.5628053E-02 | 8.7500000E-01 | 2.0214923E-01 | 4.0171812E-15 | -1.4304151E-01 | -5.9161867E-15 | 7.0821018E-02 |

| | $t^7$ | $t^8$ | $t^9$ | $t^{10}$ | $t^{11}$ | $t^{12}$ | $t^{13}$ |
|---|---|---|---|---|---|---|---|
| relu | 1.8094049E-15 | -8.5698408E-02 | 1.4490894E-17 | 1.6559778E-02 | -9.7673166E-17 | -2.1229556E-03 | 2.3344256E-17 |
| tanh | -2.9502585E-02 | 1.1578462E-15 | 5.9214502E-03 | -2.4554185E-16 | -8.5681435E-04 | 3.3129449E-17 | 8.8697276E-05 |
| softplus | -3.2764837E-15 | -2.5656019E-05 | 1.0262503E-15 | 1.9069649E-06 | -2.1426831E-16 | -1.2924529E-07 | 3.0466611E-17 |
| elu | -4.5051680E-02 | -4.3845220E-03 | 1.1333777E-02 | 7.8485900E-04 | -1.9013645E-03 | -9.6036723E-05 | 2.1786860E-04 |
| gelu | 2.0301054E-15 | -1.1630904E-03 | -1.4744780E-16 | 1.0800433E-04 | -5.4350867E-17 | -7.9444190E-06 | 1.6680823E-17 |
| htanh | 3.3738173E-01 | -1.5872645E-15 | -9.5223816E-02 | 3.6951250E-16 | 1.6867158E-02 | -5.4818441E-17 | -1.9874811E-03 |
| lrelu | -1.3304595E-15 | -8.4841423E-02 | 7.6609561E-16 | 1.6394181E-02 | -2.1268001E-16 | -2.1017260E-03 | 3.5039736E-17 |
| selu | -7.9205328E-02 | 5.2914487E-02 | 1.9925905E-02 | -1.0334500E-02 | -3.3427877E-03 | 1.3329332E-03 | 3.8303464E-04 |
| sigmoid | -2.0739824E-04 | 1.3731145E-15 | 1.9820840E-05 | -2.1665218E-16 | -1.7123664E-06 | 2.1314190E-17 | 1.2507122E-07 |
| lrelu -0.25 | 1.6962453E-15 | -6.4273806E-02 | -6.5505775E-17 | 1.2419834E-02 | -6.1909935E-17 | -1.5922167E-03 | 1.6274033E-17 |
| lrelu -0.75 | 2.8720156E-15 | -2.1424602E-02 | -7.1728555E-16 | 4.1399446E-03 | 1.0211438E-16 | -5.3073889E-04 | -8.1962396E-18 |

| | $t^{14}$ | $t^{15}$ | $t^{16}$ | $t^{17}$ | $t^{18}$ | $t^{19}$ | $t^{20}$ |
|---|---|---|---|---|---|---|---|
| relu | 1.8443579E-04 | -2.8820282E-18 | -1.0903101E-05 | 2.1693327E-19 | 4.3182633E-07 | -1.0312989E-20 | -1.0961345E-08 |
| tanh | -2.9550924E-18 | -6.5355890E-06 | 1.7658317E-19 | 3.3887465E-07 | -6.9979237E-21 | -1.2057256E-08 | 1.7656046E-22 |
| softplus | 7.3223066E-09 | -2.9626728E-18 | -3.2257985E-10 | 1.9577707E-19 | 1.0362969E-11 | -8.6141997E-21 | -2.2542826E-13 |
| elu | 8.0862081E-06 | -1.7269223E-05 | -4.6740594E-07 | 9.4564265E-07 | 1.8201823E-08 | -3.5094476E-08 | -4.5598427E-10 |
| gelu | 4.5500348E-07 | -2.2272244E-18 | -1.9681332E-08 | 1.7448146E-19 | 6.1548299E-10 | -8.5046796E-21 | -1.3032819E-11 |
| htanh | 5.3651152E-18 | 1.5990800E-04 | -3.5046880E-19 | -8.8268590E-06 | 1.5117444E-20 | 3.2893167E-07 | -4.1329037E-22 |
| lrelu | 1.8259143E-04 | -3.6811891E-18 | -1.0794070E-05 | 2.5320591E-19 | 4.2750807E-07 | -1.1364747E-20 | -1.0851732E-08 |
| selu | -1.1625322E-04 | -3.0361009E-05 | 6.8910899E-06 | 1.6625337E-06 | -2.7347262E-07 | -6.1699575E-08 | 6.9523720E-09 |
| sigmoid | -1.3305631E-18 | -7.2716001E-09 | 5.1348407E-20 | 3.1974208E-10 | -1.0979379E-21 | -1.0107329E-11 | 7.6307576E-24 |
| lrelu -0.25 | 1.3832684E-04 | -2.0580252E-18 | -8.1773260E-06 | 1.5611204E-19 | 3.2386975E-07 | -7.4361410E-21 | -8.2210090E-09 |
| lrelu -0.75 | 4.6108947E-05 | 2.9304331E-19 | -2.7257753E-06 | 6.3999589E-21 | 1.0795658E-07 | -1.1416736E-21 | -2.7403363E-09 |

| | $t^{21}$ | $t^{22}$ | $t^{23}$ | $t^{24}$ | $t^{25}$ |
|---|---|---|---|---|---|
| relu | 3.0296993E-22 | 1.6113608E-10 | -5.0295283E-24 | -1.0429164E-12 | 3.6122193E-26 |
| tanh | 2.7986355E-10 | -2.5687606E-24 | -3.8131280E-12 | 1.6410322E-26 | 2.3116680E-14 |
| softplus | 2.4105425E-22 | 2.9464505E-15 | -3.8765757E-24 | -1.7393697E-17 | 2.7255018E-26 |
| elu | 8.4227758E-10 | 6.6327810E-12 | -1.1791444E-11 | -4.2559630E-14 | 7.3104430E-14 |
| gelu | 2.5429110E-22 | 1.6629623E-13 | -4.2790160E-24 | -9.6171035E-16 | 3.1074839E-26 |
| htanh | -7.9089452E-09 | 6.4866408E-24 | 1.1077249E-10 | -4.4517362E-26 | -6.8651287E-13 |
| lrelu | 3.2080448E-22 | 1.5952472E-10 | -5.1716715E-24 | -1.0324872E-12 | 3.6319847E-26 |
| selu | 1.4808077E-09 | -1.0232631E-10 | -2.0730531E-11 | 6.6293325E-13 | 1.2852485E-13 |
| sigmoid | 2.1491988E-13 | 1.3841516E-25 | -2.7389162E-15 | -2.2476903E-27 | 1.5757713E-17 |
| lrelu -0.25 | 2.1839093E-22 | 1.2085206E-10 | -3.6215891E-24 | -7.8218726E-13 | 2.5980741E-26 |
| lrelu -0.75 | 5.0212219E-23 | 4.0284020E-11 | -1.0326312E-24 | -2.6072909E-13 | 8.4909511E-27 |

Table 1: Numerical values for the polynomial coefficients used in Figure 18 and 19.

