# OpenReview forum: "Ringing ReLUs: Harmonic Distortion Analysis of Nonlinear Feedforward Networks"
_ICLR.cc/2021/Conference — ICLR 2021 Poster_

### Official Review · AnonReviewer2 · 2020-10-28
**Initial review**

**Rating:** 8
**Confidence:** 4

**Review:**

***Summary***

I would firstly like to thank the authors for an interesting read. I enjoyed going through the submission very much.

The authors propose to understand the qualitative effects of nonlinearities by studying the impact they have on the Fourier spectrum of deep neural networks. The central hypothesis is that nonlinearities with a lot of energy in their side lobes (high frequencies), lead to neural networks that have a rougher mapping and that are consequently tougher to train because the derivative landscape is also rougher. They back this hypothesis up with some mathematical arguments from the area of harmonic distortion analysis and with empirical experiments to support the qualitative predictions of this theory.

***Pros***

I found the submission very readable. I think the balance of text to mathematics in the main submission was about right, reserving the appendix for a more in depth discussion.

I think that while the central finding that deep mappings are smoother is, in itself, not particularly novel, the chain of reasoning to get to this fact is new. I like the use of the Fourier spectrum to show this and the analysis behind how the spectra of various nonlinearities affect overall network smoothness..

The choice of experiments, which sequentially back up the claims, makes for a good paper. I particularly enjoyed the results in Figure 2, which were very instructive and gave good insight into the predictions of the theory.


***Cons and constructive feedback***

In order from start to finish.

In the abstract should differential be differentiable?

I think a good paper to cite would be “Avoiding Pathologies in Very Deep Networks” (Duvenaud et al., 2014) who analyze deep kernels in Gaussian processes. While the underlying models are different, the kinds of qualitative results in this paper are very similar to the submission.

I am concerned about the use of the Fourier spectrum to model the ReLU nonlinearity. Will there not be issues with the Gibb’s phenomenon? The discontinuous gradient will mean that a spectrum exists, but reconstructions are poor.

Paragraph below equation 3: uniformely -> uniformly

Equation 4: using t_j is confusing given that you use t in eqn 1. Please change to another symbol

Eqn 6: Please define the autocorrelation symbol in the main text.

Eqn 6: Please define z versus z_j

Section 3.2 discussion: I would assume that while higher order autocorrelations would broaden the spectrum they would also smooth it out. For high orders it would like Gaussian-like in shape. This would not necessarily lead to blue-shifting.

Section 3.3: therfore -> therefore

Section 3.4 trivial -> trivially

Section 3.5: Exponential downweighting. ResNets have combinatorially more medium length paths than short or long ones. So the average weight of a medium path is far higher than short or long ones. I would have liked to have seen a deeper analysis of this effect.

Experiments: I found these very interesting. What is the motivation for only focussing on networks at initialization? I would have loved to have seen what a pertained network looks like.

Are ensembles covered within the scope of this theory? They seem to have good performance but since each member is trained individually there is no smoothing of the training function, although the test loss function is smoother when all member models are combined.


***Post rebuttal review***

Having read the rebuttal, I am very happy with the author responses. My main concerns about the Gibb's phenomenon and the choice to consider blueshifting at initialization have been thoroughly addressed. It is clear to me that the authors have thought long and hard about the rebuttal and used it to improve their submission. Therefore I maintain that this is still a clear accept.

---

> ### Author Response · Authors · 2020-11-17
> **Author Response 2 1/2**
>
> We would like to thank Reviewer 2 for the constructive and very detailed feedback. We have uploaded a revised version of the paper that addresses the issues raised in the review, including spelling mistakes and ambiguous formulations. For the technical questions, we would like to provide answers and further explanations below:
>
> *Q: a good paper to cite would be “Avoiding Pathologies in Very Deep Networks” (Duvenaud et al., 2014)*
>
> A: Thanks for providing the reference. It is interesting to see a similar discussion for wide-limit networks. We see our paper as complementary since the authors do not discuss how the choice of nonlinearity affects the strength of the pathologies. We have added the reference to our related work section.
>
> *Q: In the abstract should differential be differentiable?*
>
> A: “differential” – we mean measures that consider derivatives of the loss function at a point (usually to assess the sharpness of a minimum). In contrast, we aim at assessing the roughness of the loss surface in a less localized fashion, considering the power spectrum of the Fourier transform to characterize the function’s behavior.
>
> *Q: I am concerned about the use of the Fourier spectrum to model the ReLU nonlinearity. Will there not be issues with the Gibb’s phenomenon? The discontinuous gradient will mean that a spectrum exists, but reconstructions are poor.*
>
> A: This needs a bit of discussion (also added to the revision): In the continuous, theoretical model, the Fourier series will converge in an L2-sense even for non-smooth nonlinearities (like ReLU) as long as the network has finite variation (which all practical networks do). This means that the subset of the domain with low approximation quality (such as overshoots from Gibb’s phenomenon) will shrink to measure zero with growing frequency order. Discontinuities (including higher-order ones) will show up as an increase in magnitude of high-frequency coefficients (as intended, these are the harmonics of such a nonlinearity; their energy reflects the amount of non-linear deviation).
>
> In the experimental part, we use the discrete Fourier transform (DFT) to compute spectra. The DFT is a unitary 1:1 map between finite-dimensional vector spaces and thus itself does not suffer from approximation quality problems for any resolution chosen (at the discretization points; we never evaluate a continuous interpolation). However, applying the DFT on samples of an originally continuous function, necessarily sampled with finite resolution, can lead to aliasing, as discussed in section 4.1 of the paper. We therefore have to use sufficient sampling to capture the high frequency content, otherwise the blue-shift effect would be underestimated (because aliasing falsely amplifies lower frequencies).
>
> *Q: Section 3.2 discussion: I would assume that while higher order autocorrelations would broaden the spectrum they would also smooth it out. For high orders it would like Gaussian-like in shape. This would not necessarily lead to blue-shifting.*
>
> A: If we assume a real positive function (with second moments) as spectrum, the repeated autoconvolution will eventually converge to a Gaussian distribution. However, the Gaussian will have a standard deviation $\sqrt{n}$ for $n$ autoconvolutions. This means that the spectrum will broaden, but less dramatically than a simpler worst-case estimate suggest (the limit would be the maximum support of $O(n)$). The spectrum is of course complex rather than positive, the analysis therefore only suggests a Gaussian weighting, not necessarily final shape (but qualitatively, we would expect a similar reduction of the broadening effect). The analogy to the central limit theorem is already discussed in Appendix B4; we will add a reference to the main paper in the revision.
>
> *Q: Section 3.5: Exponential downweighting. ResNets have combinatorially more medium length paths than short or long ones. So the average weight of a medium path is far higher than short or long ones. I would have liked to have seen a deeper analysis of this effect.*
>
> A: This is an important point (we will add a clarification in the revision). Combinatorially, a ResNet has a 50-50-weighting for each block visited, therefore halving the average path length. However, the weight of these paths is not 50-50, as stacks of non-bypass-networks suffer from vanishing gradients (Veit et al., “Residual Networks Behave Like Ensembles of Relatively Shallow Networks”). At initialization, before running any batch-normalization, this seems to be a consequence of the uneven spectrum of Gaussian matrices which are uncorrelated across multiple layers (therefore attenuating in all directions when stacked). Empirically, the dampening is maintained after training (Veit et al.). When taking the dampening effect into account, the actual weight of the residual blocks become smaller and thus the contribution to the output too (this leads to the actual exponential downweighting).

---

> > ### Author Response · Authors · 2020-11-17
> > **Author Response 2 2/2**
> >
> > *Q: What is the motivation for only focusing on networks at initialization? I would have loved to have seen what a pertained network looks like.*
> >
> > A: We decided to show the effect of blueshift in the paper around the initialization point because it is very visible there. In the appendix, Figure 10 shows that during training, the blueshift effect gets progressively weaker but is still present. Appendix Figure 11 shows that also when repeating the experiment of Figure 2 exclusively in gradient direction, blueshift is also weaker but still present. Blueshift being mostly present around initialization is not an argument for it being irrelevant in training, since early training phases are known to being fundamental(e.g.: “Critical Learning Periods in Deep Neural Networks” Achille et al.).
> >
> > *Q: Are ensembles covered within the scope of this theory? They seem to have good performance but since each member is trained individually there is no smoothing of the training function, although the test loss function is smoother when all member models are combined.*
> >
> > A: Ensemble models are explained in our theory by the FDSA (Frequency dependent signal averaging) effect. In an ensemble, we would expect the loss surfaces of the sub-networks to mostly share their coarse structure (low frequencies), while presenting differences mostly in the details (high frequencies). By consequence, the mostly uncorrelated high frequencies average out and lose relative weight and the resulting ensemble loss surface is smoother than the individual loss surfaces.

---

### Official Review · AnonReviewer4 · 2020-10-29
**An interesting measure and view of the deep neural network**

**Rating:** 5
**Confidence:** 4

**Review:**

Summary: The paper applies harmonic distortion analysis to understand the effect of nonlinearities in the spectral domain. This gives global view of the network output roughness.

Strong points: The paper introduces an interesting measure "roughness" of deep neural network via harmonic distortion analysis. It evaluates the blueshift near the initialization for various nonlinearity (ReLU, LReLU, TanH, Sigmoid,…) and architecture choices (no skip, skip, depth, width). The blueshift fits people's intuition of the architectural choices.

Weak points:
1. Current presentation mainly focus on explaining existing networks via blueshift measure. It does not "predict" new choices of the nonlinearity and architectures. This prediction could be related with the data: find the one nonlinearity and architectures that could best fit the data complexity.
2. The hypothesis that spectral blueshift impedes training fit the observations that architectural choices with  good harmonics generation control are easier to train to good performance. However this hypothesis is not consistent with the Sigmoid +NS case in Figure 16. I suppose it is hard to optimize Sigmoid+NS for a deep network (50 layers).
3. The paper claims this can give a global view of the roughness. But the measurements are near the initialization. In this sense, it would be better to look at the roughness measure out of the initialization neighborhood.

I would not recommend the acceptance for now because of the above weak points.

After the rebuttal

I thank the authors for the detailed feedback. I am a bit with AnonReviewer3 about the concerns on the descriptive languages of the paper "less nonlinear", "more expressiveness". Moreover, the behaviors of different nonlinearities in the Fig.3 and Fig.17 are related to the specific initialization and batch normalization, which is rather not a global view of the landscape. I would keep the score unchanged.

---

> ### Author Response · Authors · 2020-11-17
> **Author Response 2**
>
> We would like to thank Reviewer 4 for the the constructive feedback. We would like to address the interesting questions brought up in the review. We will also address these issues in the revised paper has been uploaded.
>
> *Q: Current presentation mainly focus on explaining existing networks via blueshift measure. It does not "predict" new choices of the nonlinearity and architectures. This prediction could be related with the data: find the one nonlinearity and architectures that could best fit the data complexity. *
>
> A: We agree with that it would be interesting to predict the optimal amount of nonlinearity needed for a certain problem (for example in Figure 6); But this would require to a-priori knowledge of the complexity of the data to fit, which is an open research question and not in the scope of the paper; our goal is only to explain how different nonlinearities and residual connections (or parallel paths) affect the shape of the loss surface.
>
> *Q: The hypothesis that spectral blueshift impedes training fit the observations that architectural choices with good harmonics generation control are easier to train to good performance. However this hypothesis is not consistent with the Sigmoid +NS case in Figure 16. I suppose it is hard to optimize Sigmoid+NS for a deep network (50 layers). *
>
> A: Yes, this is an interesting aspect to discuss further: As a concrete comparison, Sigmoid and tanh only differ in scaling (in particular along the domain axis, which makes sigmoid “less nonlinear” if the input distribution remains fixed, see Fig. 17). ReLu would be even “more nonlinear” in our model. In practice, sigmoid does indeed reach a worse performance than tanh or relu when used as a drop-in replacement, but this is not inconsistent with our theory: since sigmoid is a more linear function than tanh, the expressivity of the network is worse and so the total performance of the network might suffer. Our theory only predicts that the time to reach the network’s final performance is not impeded by performing gradient descent on a rougher loss surface. Indeed, we conducted an experiment with the hyperparameters from the paper on a ResNet56 “NoShort” and when superposing the training curves, the training speeds of sigmoid, tanh and relu appear comparable (sigmoid is even a bit faster at the beginning of the training where blueshift is stronger), but the final training accuracy reached increases quite a bit with nonlinearity (best for ReLu, second for tanh, least for sigmoid). We added this plot the “rebuttal” folder in the additional material.
>
> *Q: The paper claims this can give a global view of the roughness. But the measurements are near the initialization. In this sense, it would be better to look at the roughness measure out of the initialization neighborhood.*
>
> A: We perform the experiments near the initialization point as the initial behavior of the function appears to have particular impact on the results (see “Critical Learning Periods in Deep Neural Networks”, Achille et al.). The paper includes experiments during and after training (Fig. 10), as well as measurements in gradient directions only (Fig. 11) in the appendix. The experiments show that the blueshift effects becomes weaker during training but still remains present. See also the corresponding answer to Rev. 2.
>
> The results are more global than a differential analysis that looks only at a gradient at single points. The Fourier view shows the norm of the gradient function over a finite-length path (or averages of many such path). It is true (and important to stress) that the view is not fully global - the domain of the network is unbounded and infinite, with common nonlinearities saturating in the outer regions. Therefore a spatial restriction is required (our experiments sample path starting from the current state of the network, initialization or later [appendix], in random directions, but keep the path length short to remain representative of that region).
>
> We agree that this could be discussed more clearly; we will add qualifications to the revised version.

---

### Official Review · AnonReviewer3 · 2020-10-29

**Rating:** 4
**Confidence:** 4

**Review:**

Summary: This paper proposes a new approach for how to analyze the ruggedness of the surface of the neural network loss. Specifically, the paper proposes to apply harmonic distortion on the weight-to-output (w-o) maps. That is, the method casts the w-o functions in the Fourier domain and then aggregate the surface characteristics by virtue of averaging the different order Fourier coefficients. The  paper shows that non-linearities are responsible for blueshifting with deeper layers, that is for "transferring more energy" on the higher frequencies. The consequence is rougher surfaces, as well as higher frequencies for gradients, which can lead to exploding gradients in the deeper layers. The remedy is with skip connections and feature averaging, which although are methods already known to improve trainability, the paper corroborates that they also make sense in terms of said approach. The paper conducts various empirical and ablation studies, providing evidence of the claims.

The strengths of the paper:
- I believe that at the core the paper offers some really nice preliminary ideas and intuitions. It is very intuitive that deeper layers generate higher frequencies to the loss and thus can make the optimization harder. On the other hand, higher frequencies are necessary for expressivity. This is another instantiation of the bias-variance tradeoff. By finding ways to balance the relative strength of high and low frequencies, one could throttle how much expressivity is necessary for the task at hand.
- The paper goes to great lengths to support some of the claims empirically. There is lots of different experiments and it seems the claims are overall supported by the findings. Different nonlinaerities and architectures (maybe too much, in that there is less focus) are explored, which is admirable.
- I like the quality of the visualizations. It is clear that the authors have spent quite some time in generating their figures.

The weaknesses of the paper:

- Sure, if we focus only on the low-order frequencies trainability is better. Also, I dare say that the insight is not exactly surprising although very intriguing. However, what exactly is the message? That we should have only low frequencies? Or that we should have some high frequencies? That skip connections are good for better training? I believe that in many ways, the message is incomplete if one leaves out expressivity and it would be nice to extend the theory to say something about the potency of the neural network on learning patterns and generalizing. The authors already comment on this in the 'future work' lines. I think that this should become current work, otherwise the work is incomplete, at least from the current perspective.
- I have the feeling there are places where the analysis is imprecise, although it could be that I also misunderstood.
 - For one, the crux of the analysis is that the neural network nonlinearities are expressed in Fourier series (sec 3.2). Then, in the next section 3.3 the paper says that in practice nonlinearities are not polynomial and might not have a convergent Taylor expansion. So, instead a Chebyshev approximation is opted for. However, it is not clear if the Chebyshev approximation suffices or what are the limits of it? but I think this must be elaborated further.
 - Also, what are these Chebyshev approximations per nonlinearity? I think it is quite important to clarify this, considering there are nonlinearities that all but very similar, e.g., the ReLU and the leaky ReLU. What is the big difference between the two in terms of the described analysis?
- I find it hard to understand often what the analysis tries to say, either the analysis is incomplete, the writing generally unclear or I simply don't understand some details. I list my comments by order of reading (not importance).
 - Throughout the paper there is a clear desire to connect rougness with layer depth. However, in all equations and analysis the depth is not explicitly present. For instance, in equations 4-6 there is only the degree of the polynomial K, but no layer variable or index. From what I gather, the (implicit) argument is that by the successive stacking of layers, the corresponding low/high order frequencies get stronger or weaker, relatively. Then, the objective is to compare the corresponding low and high frequencies for different layers, showing that for deeper layers the higher frequencies get stronger because of the recursion. This is how depth is 'qualitatively' introduced as a variable. Is this indeed the intention? If yes, I think it can be written more explicitly.
- I find figure 1 a bit perplexing. Again, I understand what is the message, but it is hard for me to connect it to the theory, since the theory makes only indirect references to the specific nonlinearities. Also, what is 'ReLU->ResNet' supposed to stand for? ResNet is a ReLU when including skip connections? And what is 'Linear->ReLU'? To put it otherwise, adding skip connections or a ReLU nonlinearity are discrete design choices. However, the figure has continuous axis. So, what exactly is illustrated? The 'vertical' axis corresponds to the t variable in the Fourier coefficient. What about the other axis?
 - The related work points to Li et al and their spectral analysis to ground the proposed research. However, it is not explained what these observations are and how they relate to the current paper. It would be nice for the reader to add a short explanation.
 - Do we expect a difference by considering 1D slides, instead of 2D slides as motivated by Li et al? Why yes, why no?
 - It is not explained why are the mean path are empirically zero-functions. I infer that this is the case because at any location of the loss surface, if we take a small ball around it there will be an equal amount of parameters for which there is a higher or lower loss value? However, wouldn't this imply already a strong gradient (about 1, if I am not mistaken)?
 - What I find a bit confusing is that in equation 4 and 5 we apply the nonlinearity \phi on p(t). However, p(t) are the 1-D slices of our neural network f defined in the preamble of section 3. I would assume that the nonlinearities would then already be inside p(t). In fact, in the preamble of 3 there is also a mention of \phi and how p(t) is a polynomial when \phi is the identity function. Maybe I have misunderstood something here.
 - There is an attempt to connect exploding gradients to blueshifting. However, this is not entirely clear to me. Indeed, one can say that simultaneously we have blueshifting in the gradient and at the same time exploding gradients. Does this mean that one cause the other, however? Couldn't one have explosions by having disproportionately large low order frequencies (not that it is the case, just wondering)? Or some other phenomenon.
 - There is a connection to exploding gradients, however, in deep networks vanishing gradients also important (maybe more so). Can the analysis address also vanishing gradients?
 - It is not exactly clear why the frequency dependent signal averaging is wekaner than exponetial downweights. The explanation is very brief and a bit vague (law of large numbers, exponential decay). Is there a more precise qualitative or quantitative argument here?
 Can one still call the method as more 'global' in saying something about roughness, given that all coefficients are computed per layer? Of course, each layer's coefficient are influenced by all previous layers, but is this enough to paint the method 'global'?
 - Is somehow K (polynomial order) connected to L (number of layers)? Or is this relation the way I described above?
 - Perhaps relevant to the previous point, and taking the position of the devil's advocate, in a way what is put forward by the paper is a re-interpretation of existing knowledge. While certainly very intriguing, is there a new insight on trainability for a new type of method/technique that can improve trainability? What about wide layers and them being easier/harder to train?
 - The text in p. 6 on Fig 5 (In figure 5, we use the power law ... ) is unclear to me.
 - In p. 7 there is a great misalignment between the figure references and the figure locations in the paper.
 - How is leaky ReLU connected to other nonlinearities? How precisely does it make a difference? How difference is the Chebyshev polynomial?
 - What does it mean 'making the networking more linear'? Do you mean increasing the \alpha hyperparameter till it becomes 1, in which case you have a linear function?

In general, I find the paper quite interesting and with valuable potential contributions, but incomplete and not ready for publication at this stage. I believe it would worth it if the authors took the time to revisit the crispness of the message as well as the writing. Of course, I am more than happy to revisit my recommendation if the authors produce a convincing argument.

---

> ### Author Response · Authors · 2020-11-17
> **Author Response 3 1/3**
>
> We would like to thank Reviewer 3 for the great amount of helpful feedback. A revised version of the paper that fixes typos and figure misplacements and clarifies the discussion has been uploaded. We would now like to answer the open technical questions:
>
> *Q: [...] However, what exactly is the message? That we should have only low frequencies? Or that we should have some high frequencies? [...]*
>
> A: Yes, we agree that non-linearity (which shows up as harmonics in the spectrum) is required for expressive networks. At the same time, this might also impede training. Fig. 6 probably illustrates this trade-off quite well. We believe it also shows that ResNets are more successful in trading-off these effects, and we do not fully understand why they remain reasonably expressive. A certain amount of nonlinearity is needed in a network – how to distribute it well in a network (over layers and training time), is an open research question.
> The message (main contribution) of the paper is to understand the effect of  stacks of nonlinearities and architectural features such as shortcuts or parallel computation paths on the shape (roughness) of the loss surface better by looking at harmonic generation in the Fourier domain. This is consistent with previous observations (see also the reply to R1), but puts these in a new, common framework. We consider this more of a tool for future research than an answer to the hard question of finding the optimal trade-off (which likely would require understanding better which kind of “expressivity” is most important to typical data, which appears to be a very hard to answer question). We updated section 5 for improved clarity.
>
> *Q: Throughout the paper there is a clear desire to connect roughness with layer depth. However, in all equations and analysis the depth is not explicitly present. For instance, in equations 4-6 there is only the degree of the polynomial K, but no layer variable or index. From what I gather, the (implicit) argument is that by the successive stacking of layers, the corresponding low/high order frequencies get stronger or weaker, relatively. Then, the objective is to compare the corresponding low and high frequencies for different layers, showing that for deeper layers the higher frequencies get stronger because of the recursion. This is how depth is 'qualitatively' introduced as a variable. Is this indeed the intention? If yes, I think it can be written more explicitly.*
>
> A: Thanks for the comment. Yes, the analysis has recursive structure. Upon rereading the paper afterwards, we noticed that the original text did not convey this clearly, and we have added clarifications in the revised version. In our model, harmonic distortion analysis is applied to a single nonlinearity, which is fed a signal p(t) created by the previous layers. The incoming spectrum will be broadened by the autoconvolutions, and a repeated application of this process will lead to increasing blue-shifting. Therefore, we can assume a constant degree K of the nonlinearities and there is no explicit layer index; everything only applies to one layer in that equation.
>
> *Q: For one, the crux of the analysis is that the neural network nonlinearities are expressed in Fourier series (sec 3.2). Then, in the next section 3.3 the paper says that in practice nonlinearities are not polynomial and might not have a convergent Taylor expansion. So, instead a Chebyshev approximation is opted for. However, it is not clear if the Chebyshev approximation suffices or what are the limits of it? but I think this must be elaborated further.*
>
> A: Our current theoretical analysis only holds for polynomial nonlinearities. It shows that larger non-linearity in the sense of larger higher-order polynomial coefficients lead to more blueshift. The Stone-Weierstrass theorem would of course permit an approximate of continuous function (as output by all relevant nonlinearities) as closely as desired by a polynomial, in order to consider a wider range of functions. However, the effect on the spectrum still remains harder to formally establish, as we have to consider two limit processes which cannot trivially be exchanged. While we are not able to formally prove this, it is still reasonable to conjecture that a tight approximation with finite degree (Figure 17) is sufficient to predict blueshifts of non-polynomial nonlinearities. We verify this experimentally by relating the drop-off rate of the polynomials obtained (Figure 18) with measured blueshift (Figure 16), which yields qualitatively correct results. We use a least-squares Chebyshev approximation for this experiment because it is easy to compute and (reasonably) stable for larger degrees (unlike, for example, equidistant point-wise fits, which oscillate, or Taylor expansions, which might not convergence).

---

> > ### Author Response · Authors · 2020-11-17
> > **Author Response 3 2/3**
> >
> >
> > *Q: I find figure 1 a bit perplexing. Again, I understand what is the message, but it is hard for me to connect it to the theory, since the theory makes only indirect references to the specific nonlinearities. Also, what is 'ReLU->ResNet' supposed to stand for? ResNet is a ReLU when including skip connections? And what is 'Linear->ReLU'? To put it otherwise, adding skip connections or a ReLU nonlinearity are discrete design choices. However, the figure has continuous axis. So, what exactly is illustrated? The 'vertical' axis corresponds to the t variable in the Fourier coefficient. What about the other axis?*
> >
> > A: Figure 1 shows two transitions: First, we tune the “alpha” parameter of Leaky ReLU to continuously transition from a linear network ($\alpha=1$ means no non-linearity) to a standard ReLU network. The network used is a ResNet56, but we set multiply the shortcut connections with an additional scalar, which we set to zero for this part (so there are no shortcuts). In the second transition, we increase the shortcut weight $\nu$ from zero to one, transitioning continuously from a ReLU-network without shortcuts to a ResNet. The upward axis is the network output, and $t \in [0,1]$ is a parameter to move along a fixed random path. We have reworded section 4.4 clarifying this.
> >
> > *Q: [...] Li et al. ... It would be nice for the reader to add a short explanation.*
> >
> > A: We updated the related work, explicitely stating what results of Li et al. were built on and including the referenced paper.
> >
> >
> > *Q: Do we expect a difference by considering 1D slides, instead of 2D slides*
> >
> > A: We do not expect a big difference comparing 1D slices to 2D slices because empirically the loss surface is looks rather very isotropic in random directions.
> >
> > *Q: It is not explained why are the mean path are empirically zero-functions. [...]*
> >
> > A: We have measured sample path and consistently observed mean Fourier coefficients near zero (excluding the DC coefficient $z_0$, which does have a non-vanishing expectation). As the variance (power spectrum) is large, an average path will still show gradients in various directions (which, to reach zero expectation approximately average out to zero on hyperspheres of constant distance to the starting point). The reason is indeed that we have selected the path length to be small and thus do not expect a significant up or down trend near a random initialization point. This does no longer hold if we consider the *loss surface* and at or near a local minima since the loss tends will go up in every direction or if we consider loss anywhere at very large path length. This bias seems to disappear again when observing neural output instead of the loss function (which is the case in most visualizations) since the output of single neurons is not minimized. We added a more detailed discussion to the revision.
> >
> > *Q: There is a connection to exploding gradients [...] Can the analysis address also vanishing gradients?*
> >
> > A: We have updated section 3.4 to clarify that blueshift generally causes exploding gradients (blueshift denotes harmonic creation at non-linearities; thus, applications of nonlinearities will increase frequency and thus gradient magnitude, see below). It is worth noting that the model predicts that the the *lowest* layers (close to the input) will have the largest gradient magnitude increase. Gradient magnitude shrinks with increasing layer index. Relatively speaking, one could also call this effect vanishing gradients.

---

> > > ### Author Response · Authors · 2020-11-17
> > > **Author Response 3 3/3**
> > >
> > > *Q: There is an attempt to connect exploding gradients to blueshifting. However, this is not entirely clear to me. Indeed, one can say that simultaneously we have blueshifting in the gradient and at the same time exploding gradients. Does this mean that one cause the other, however? Could n't one have explosions by having disproportionately large low order frequencies (not that it is the case, just wondering)? Or some other phenomenon.*
> > >
> > > A: If z_k are the complex Fourier coefficients of a (path-) function p(t), then -i*k*z_k are the coefficients of its derivative function p’(t). In other words, taking derivatives is a high-pass filter in Fourier domain, amplifying coefficients linearly with frequency. This connects blue-shift-effects to exploding gradients: The norm of the gradient function is (by Parsevals theorem)  $\sqrt(\sum_k(k^2 \cdot z_k^2))$. Thus, the norm of the gradient function increases when “energy” in the spectrum shifts from lower to higher frequencies, even if the magnitude of the coefficients does not change. Then, if the function norm increases, there must be on average gradients of large magnitude (somewhere) along the path (in the original, untransformed domain) in order to integrate to an overall larger norm.
> > >
> > > Exploding gradients can of course also be caused by other phenomena unrelated to blueshift, e.g. by stacking weight matrices with a non-uniform singular spectrum (as shown in “Exact solutions to the nonlinear dynamics of learning in deep linear neural networks” by Saxe et al. in the complete absense of nonlinearities). We clarified this in the text.
> > >
> > > *Q: It is not exactly clear why the frequency dependent signal averaging is wekaner than exponetial downweights. The explanation is very brief and a bit vague (law of large numbers, exponential decay). Is there a more precise qualitative or quantitative argument here? Can one still call the method as more 'global' in saying something about roughness, given that all coefficients are computed per layer? Of course, each layer's coefficient are influenced by all previous layers, but is this enough to paint the method 'global'?*
> > >
> > > A: In a residual network, the weight of the residual branch is lower than that of the bypass because of vanishing gradient effects (for example, at initialization the weight matrices are Gaussian matrices, which are known to have a non-uniform singular value spectrum; stacking several independently chosen matrices will dampen the signal in all directions; see also the corresponding answer to R2). Hence, if we consider “paths” through multiple non-linear non-shortcut  blocks, the weight decreases exponentially with their number (see Veit et al., “Residual Networks Behave Like Ensembles of Relatively Shallow Networks”, Fig. 6c). In contrast, averaging independently computed signals will have an uncorrelated blueshift spectrum, which thus will average out to zero according to the law of large numbers. The convergence rate here is only $n^{(-1/2)}$, which is much slower.
> > >
> > > *Q: How is leaky ReLU connected to other nonlinearities? How precisely does it make a difference? How difference is the Chebyshev polynomial?
> > > Q: What does it mean 'making the networking more linear'? Do you mean increasing the \alpha hyperparameter till it becomes 1, in which case you have a linear function?*
> > >
> > > A: We use Leaky ReLU to control the amount of nonlinearity in the network ( "more linear" for alpha -> 1). If we approximate the leaky ReLU function by polynomials (using Chebyshef fits of degree 20) for varying alpha (see Figure 18d/e in the appendix), we can see how this leads to a slower drop-off the closer we get to ReLu from linear.

---

### Official Review · AnonReviewer1 · 2020-10-30
**Hanrmonic distorsion in deep neural networks**

**Rating:** 8
**Confidence:** 3

**Review:**

The papers proposes an interesting analysis that links several aspects of architectural design in Deep NNs to the spectral analysis and observed roughness. Different activations functions are considered in the study, mainly centered on deep CNN with or without skip connections (in the framework of ResNet v1 and v2). The starting point, which is not novel, actually, but relevant, is that specific types of non-linearities introduce harmonic distortions, and the effect is potentially amplified when multiple non-linearities are stacked. Theoretically, the paper shows that there is a concrete link between architectural choices in the network design and the blueshift in the frequency domain. Experimentally, the observations support the mathematical analysis. All in all, some of the conclusions regarding trainability of CNN architectures with skip connections have been already noted and do not seem greatly new, but the paper introduces a nice perspective to see this phenomenon in another light.
The paper is generally well written and I appreciated reading it.

The major downside I see in the current form of the manuscript is given by some aspects of the presentation. For instance, Fig. 1 is clearly misplaced (it should be in Section 4.4). Similarly, Figure 6 should be in Section 5. Moreover, abbreviations would be better used in a more uniform manner (e.g., SDFA, FDSA, SDSA). Regarding the reported experiments: are the given plots achieved by averaging over multiple runs? (only in one of the many experimental settings this information is given in the paper). Finally, the link between the left and the right sides in Figure 6 is not really straightforward, perhaps grouping together the short and the noshort results could be of help for the reader.

-- EDIT:
Thanks for the nice feedback during the rebuttal. I am happy to stay with my rating of clear acceptance.

---

> ### Author Response · Authors · 2020-11-17
> **Author Response 1**
>
> We would like to thank Reviewer 1 for the constructive feedback. We have uploaded a revised version of the paper with improved figure placement and that fixes the misnomers pointed out. We will also update Figure 6 which was admittedly botched for a lack of space to include v2 blocks as well. Concerning the overall goal, we agree with the assessment: Indeed, the intention of this paper is to gain a new perspective on already known phenomena, namely the impact of the choice of the nonlinearity and architecture on the degradation of the output surface with regard to network depth.
>
> *Q: Are the given plots achieved by averaging over multiple runs?*
>
> A: Only the training plots contain multiple runs (5 runs for Figure 8 and 1 run per cell in Figure 6) as the blueshift plots are already averaged over many neurons/paths/initializations. We clarified this in the revised version of the paper.

---

### Decision · Program_Chairs · 2021-01-07
**Final Decision**

**Decision:**

Accept (Poster)

**Comment:**

The paper presents an analysis of the spectral impact of non-linearities in a neural network, using harmonic distortion analysis as a means to quantify the effect they have in the spectral domain, linking a blue-shift phenomenon to architectural choices. This is an interesting analysis, that could be strengthened by a more thorough exploration of how this analysis relates to other properties, such as generalization, as well as through the impact of the blueshift effect through the training process.